# Causal network inference from gene transcriptional time-series response to glucocorticoids

Jonathan Lu [1], Bianca Dumitrascu [2], Ian C. McDowell [3], Brian Jo [2], Alejandro Barrera [4,5], Linda K. Hong [4], Sarah M. Leichter [4], Timothy E. Reddy [6]*, Barbara E. Engelhardt [1,7]*

**1** Department of Computer Science, Princeton University, Princeton, New Jersey, United States of America, **2** Lewis-Sigler Institute for Integrative Genomics, Princeton University, Princeton, New Jersey, United States of America, **3** Element Genomics, A UCB Company, Durham, North Carolina, United States of America, **4** Center for Genomic and Computational Biology, Duke University, Durham, North Carolina, United States of America, **5** Department of Biostatistics and Bioinformatics, Duke University Medical Center, Durham, North Carolina, United States of America, **6** Department of Genome Sciences, Duke University, Durham, North Carolina, United States of America, **7** Center for Statistics and Machine Learning, Princeton University, Princeton, New Jersey, United States of America

☯ These authors contributed equally to this work.
* tim.reddy@duke.edu (TER); bee@princeton.edu (BEE)

**Data Availability Statement:** All files have been submitted to the Gene Expression Omnibus, accession number: GSE91208. https://www.ncbi.nlm.nih.gov/geo/query/acc.cgi?acc=GSE91208 All

## Abstract

Gene regulatory network inference is essential to uncover complex relationships among gene pathways and inform downstream experiments, ultimately enabling regulatory network re-engineering. Network inference from transcriptional time-series data requires accurate, interpretable, and efficient determination of causal relationships among thousands of genes. Here, we develop Bootstrap Elastic net regression from Time Series (BETS), a statistical framework based on Granger causality for the recovery of a directed gene network from transcriptional time-series data. BETS uses elastic net regression and stability selection from bootstrapped samples to infer causal relationships among genes. BETS is highly parallelized, enabling efficient analysis of large transcriptional data sets. We show competitive accuracy on a community benchmark, the DREAM4 100-gene network inference challenge, where BETS is one of the fastest among methods of similar performance and additionally infers whether causal effects are activating or inhibitory. We apply BETS to transcriptional time-series data of differentially-expressed genes from A549 cells exposed to glucocorticoids over a period of 12 hours. We identify a network of 2768 genes and 31,945 directed edges (FDR ≤ 0.2). We validate inferred causal network edges using two external data sources: Overexpression experiments on the same glucocorticoid system, and genetic variants associated with inferred edges in primary lung tissue in the Genotype-Tissue Expression (GTEx) v6 project. BETS is available as an open source software package at https://github.com/lujonathanh/BETS.

other data are contained in the manuscript and the Supporting information.

**Funding:** This work was funded by the following grants to BEE: NIH R01 HL133218 and NIH U01 HG007900 (National Human Genome Research Institute), and an NSF 711 CAREER 1750729 (Division of Information and Intelligent Systems). The funders had no role in study design, data collection and analysis, decision to publish, or preparation of the manuscript.

**Competing interests:** I have read the journal's policy and the authors of this manuscript have the following competing interests: BEE is on the SAB for Freenome, Celsius Therapeutics, and Creyon Bio; is a consultant for Freenome; and was an employee of Genomics plc during a year of absence from Princeton University.

## Author summary

We can better understand human health and disease by studying the state of cells and how environmental dysregulation affects cell state. Cellular assays, when collected across time, can show us how genes in cells respond to stimuli. These time-series assays provide an opportunity to identify causal relationships among thousands of genes without performing hundreds of thousands of experiments. However, inferring causal relationships from these time-series data needs to be fast, robust, and accurate. We present a method, BETS, that infers causal gene networks from gene expression time series. BETS runs quickly because it is parallelized, allowing even data sets with thousands of genes to be analyzed. We demonstrate the performance of BETS compared to 22 other state-of-the-art inference methods on benchmark data. We then use BETS to build causal networks from gene expression responses to the widely-prescribed drug dexamethasone. We replicate the estimated causal relationships using gene expression data from the Genotype-Tissue Expression (GTEx) project and from additional experiments with dexamethasone. We release our software so that BETS can be used to accurately and effectively infer causal relationships from gene expression time-series assays.

This is a *PLOS Computational Biology* Methods paper.

## 1 Introduction

The recent availability of gene expression measurements over time has enabled the search for interpretable statistical models of gene regulatory dynamics [1]. These time-series data present a unique opportunity to use the coordinated transcriptional response to environmental exposure to infer causal relationships between genes. However, there are several challenges to overcome in the analysis of time-series transcriptomic data. These data are generally high-dimensional: the number of quantified gene transcripts—approximately 20,000 in human samples—often dramatically exceeds the number of available time points and samples. Many classical statistical assumptions fail to hold in this high-dimensional regime [2, 3]. Moreover, the large number of gene transcripts poses a computational burden, as the number of possible edges in a gene network grows quadratically. Also, a transcriptional time series often has a small number of time points, and those time points are often not uniformly spaced; furthermore, because transcriptional time-series data often quantify transcription post exposure, the time series is not stationary, and genes respond to the exposure and return to baseline at different rates [4, 5].

In this work, we develop an approach that uses gene transcription time series following glucocorticoid (GC) exposure to build a directed gene network [6]. GCs play an essential role in regulating stress response, and are widely used as anti-inflammatory and immunosuppresive medications [6, 7]. Dexamethasone and other GCs have recently been recommended by the U.K. Government, U.S. National Institutes of Health, World Health Organization, and Infectious Disease Society of America for treatment of hospitalized COVID-19 patients with severe disease who are on mechanical ventilation or extracorporeal membrane oxygenation [8–11]. Despite clinical benefits, prolonged exposure to GCs has been linked to increased risk of type 2 diabetes mellitus (T2DM) [12] and obesity [13]. Understanding the immune, metabolic, and transcriptional effects of GCs may enable the development of improved anti-inflammatory treatments without metabolic side effects. A recent study assayed A549 lung cells over 12 hours to characterize the effect of GCs on cell state [6]. Here, we develop a method to

accurately, interpretably, and efficiently infer a directed gene network using the study's transcription time-series data. We focus our network analysis on immune-related genes, metabolism-related genes, and transcription factors (TFs) to study the inferred coordinated response of these systems to GCs.

Our method, Bootstrap Elastic net inference from Time Series (BETS), uses vector autoregression with elastic net regularization to infer directed edges between genes. Stability selection, which assesses the robustness of an edge to perturbations in the data, leads to improvements over baseline vector autoregression methods in this high-dimensional context [3]. Furthermore, BETS is biologically interpretable because estimated coefficients provide the direction (sign) and effect size of the causal relationship between a pair of genes. Finally, BETS's parallelization enables efficient inference of networks with millions of possible edges in a computationally tractable way.

We use the causal network inferred by BETS on the GC time-series data to study the relationships between TFs, immune genes, and metabolic genes. We validate our network using two approaches: Ten measurements of the same GC system with an overexpressed TF, and an expression quantitative trait loci (eQTL) study in human primary lung tissue [14]. Although our framework is motivated by transcriptional response to GC exposure, our approaches are general, and BETS is able to infer directed networks from arbitrary high-dimensional time-series data.

## 1.1 Related work

Several methods have been developed to estimate directed gene networks from transcription time-series data (S1 Fig) [15–23]. These methods estimate directed networks in which the directed edges between nodes—representing genes—indicate a cause-effect relationship between genes. In other words, perturbing expression of the *causal gene* would lead to changes in expression of the *effect gene* [24]. We briefly overview these methods; for detailed discussion, see the Supplemental Information. Here, we take $g'$ to be the causal gene and $g$ to be the effect gene, and we quantify support for a causal edge $g' \rightarrow g$ in the time-series data.

Mutual information (MI) methods assess the MI between the expression of $g'$ at the previous time point and the expression of $g$ at the current time point (S1(A) Fig) [25–30]. A causal edge $g' \rightarrow g$ is included in the network if the MI of the two genes across time exceeds a threshold.

Granger causality methods determine if including the expression of $g'$ at the previous time point improves our ability to predict the expression of $g$ at the current time point beyond using the expression of $g$ at the previous time point [31]. A common way to implement Granger causality is through a vector autoregression (VAR) model, which assumes a linear relationship between all genes' expression at the previous time point and the expression of $g$ at the current time point. A causal edge $g' \rightarrow g$ is included in the network when $g'$ has a statistically significant coefficient in the VAR.

Ordinary differential equations (ODEs) fit the derivative of the expression of $g$ as a function of all genes' expression at a single time point (S1(C) Fig) [15, 32, 33]. ODE methods typically assume linearity, as small sample sizes make it challenging to infer the parameters of nonlinear functions. A causal edge $g' \rightarrow g$ is included when $g'$ has a statistically significant coefficient in the ODE.

Decision trees (DTs) are a type of nonparametric function based on partitioning the data [34, 35]. DT methods fall either under VAR or ODE; either the DTs fit the expression of $g$ at the current time as a function of all genes' expression at the previous time point (VAR), or they fit the derivative of the expression of $g$ as a function of all genes' expression at a single

time point (ODE) (S1(D) Fig) [36, 37]. A causal edge $g' \rightarrow g$ is included in the network when an importance score for $g'$ exceeds some threshold, where importance scores are typically the reduction in variance of $g$ when $g'$ is included as a predictor.

Dynamic Bayesian networks (DBNs) search the space of possible directed acyclic graphs between previous and current expression levels to identify the network structure with the highest posterior probability of each edge given the data (S1(E) Fig) [38–42]. DBNs typically assume a linear relationship between previous and current expression. A causal edge $g' \rightarrow g$ is included in the network when its marginal posterior probability of existence exceeds some threshold.

A Gaussian process (GP) is a distribution over continuous, nonlinear functions. GPs are often used in the context of nonlinear DBNs, where GP regression is used to model a nonlinear relationship between previous expression and current expression (S1(F) Fig) [43, 44]. A causal edge $g' \rightarrow g$ is included in the network when its posterior probability of existence exceeds some threshold.

While these approaches produce directed networks that have the flavor of Bayesian networks, except for DBNs, none of them produce graphs that are constrained to be acyclic, so they do not have the same statistical semantics as Bayesian networks.

## 2 Results

First, we briefly describe the approach that BETS uses to infer a directed gene network. Next, we compare results from BETS to those from twenty two other methods on the 100-gene time-series data from the DREAM4 Network Inference Challenge [45]. Then, we describe the network estimated from the GC transcription time-series data. Finally, we validate the inferred network using two different frameworks: Overexpression experiments on the same system, and genetic variants associated with inferred edges in human primary lung tissue in the Genotype-Tissue Expression (GTEx) v6 project [14].

### 2.1 BETS: A vector autoregressive approach to causal inference of gene regulatory networks

Directed networks represent causal relationships among diverse interacting variables in complex systems. We developed a robust, scalable approach based on ideas from Granger causality to construct these directed networks from short, high-dimensional time-series observations of gene expression levels.

Let $G$ be the set of all $p = |G|$ genes in the data set and $g \in G$ be a gene. Let $\neg g$ be $G$ with $g$ removed. Let $t$ be a single time point, ranging from $\{1, 2, \ldots, T\}$. Let $X_t^g$ be the expression of gene $g$ at time $t$. Let $L$ be the time lag, or the number of previous time point observations; so $L = 2$ means that we use two previous time points, $t - 1$ and $t - 2$, to predict expression at time $t$. These types of autoregressive models work best with similarly-spaced time points, as the data sets in this paper approximate, and assume stationarity, or the same causal effects across each time gap.

**Definition 2.1** (Granger causality). For lag $L$, a gene $g'$ is said to *Granger-cause* another gene $g$ if using $X_{t-1}^{g'}, \ldots, X_{t-L}^{g'}$, the expression values of $g'$ at times $t - 1$ to $t - L$, improves prediction of $X_t^g$, the expression value of $g$ at time $t$, beyond predicting $X_t^g$ using $X_{t-1}^g, \ldots, X_{t-L}^g$ alone.

To test for Granger causality from $g'$ to $g$, we first preprocessed the gene expression time-series data (Methods). For every potential effect gene $g$, we fit all other genes $g' \in \neg g$ simultaneously (Eq 1), echoing ideas from the graphical lasso for undirected network inference [46]. Intuitively, this adapts the idea of Granger causality to conditional Granger causality, where

we consider how gene $g'$ Granger-causes $g$ conditioning on the effects of all other genes. This approach uses the regression:

$$X_t^g = \sum_{\ell=1}^{L} \alpha_\ell^g X_{t-\ell}^g + \sum_{g' \in \neg g} \sum_{\ell=1}^{L} \beta_\ell^{g',g} X_{t-\ell}^{g'} + \epsilon_t, \tag{1}$$

where $\epsilon_t \sim \mathcal{N}(0, 1)$. For BETS, we set $L = 2$. To test for an edge, if there is statistical support for $\beta_\ell^{g',g} \neq 0$, then we say $g'$ conditionally Granger-causes $g$ at lag $L$. We build the directed network by including a directed edge to $g$ from every gene $g'$ that has been inferred to conditionally Granger-cause $g$.

Robustly building this network is difficult due to the high dimensionality of the problem: The number of genes that could Granger-cause a given $g$ far exceeds the available time points and technical replicates. To address this challenge, BETS regularizes the VAR model parameters using an *elastic net* penalty (Methods, Fig 1A). Elastic net regression encourages sparsity and performs automatic variable selection on the genes being tested for causal influence [47]. The elastic net penalty, unlike the lasso penalty [48], is able to select groups of correlated variables and allows the number of selected variables to be greater than the number of samples. This is important for gene expression assays where gene expression levels are often well correlated, and there are far more genes than samples [2].

In BETS, we fit the same VAR model to a data set in which causal genes have their expression permuted over time to generate a null distribution of edge coefficients. The coefficients

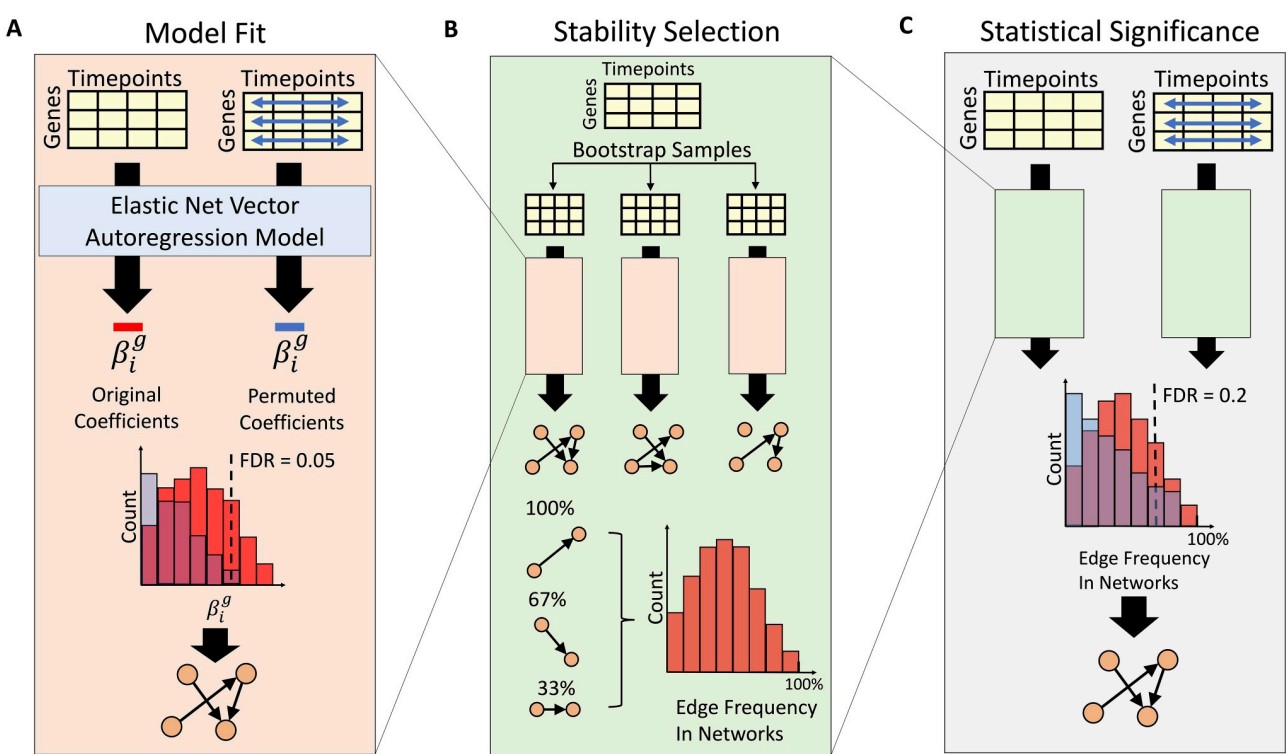

**Fig 1. BETS Algorithm. A) Model fit**. The VAR model is fit on both the original and a permuted data set (blue arrows indicate shuffling each gene's expression independently across time). Based on the null distribution of coefficients, a threshold is chosen to control the edge FDR at $\leq 0.05$. **B) Stability selection**. From the original data, 1000 bootstrap samples are generated. For each sample, a network is inferred as in A. Each edge's selection frequency across the bootstrapped networks is computed. **C) Statistical significance**. For both the original and permuted data, a selection frequency distribution is generated for stability selection as in B. Edges are thresholded to control the stability FDR at $\leq 0.2$. See S1 Fig for an overview of network inference methods.

are thresholded to produce a causal network with each edge at edge false discovery rate (FDR) ≤ 0.05 (Fig 1A). We then applied this network inference procedure to multiple (here, 1000) bootstrapped samples of the original data set (Fig 1B). Each edge has a *selection frequency*, or the frequency that the edge appears in networks inferred from the bootstrapped samples. Inspired by stability selection, this approach assesses if network edges are robust to perturbations of the data [3]. Finally, we ran this overall procedure on a permuted version of the original data set to obtain a null distribution of selection frequencies (Fig 1C). The selection frequency threshold for including each edge is chosen to control the stability FDR ≤0.2. As a baseline, we compare BETS against Enet, which runs elastic net regression without stability selection to produce a causal network with each edge at edge FDR ≤ 0.05 (Fig 1A).

## 2.2 Leading performance on DREAM Network Inference Challenge

We evaluated BETS against other directed network inference methods. We used the DREAM4 Network Inference Challenge [45], a community benchmark for directed network inference using gene time-series data. This benchmark consisted of five data sets, each with ten time-series measurements for 100 genes across 21 time points [45]. Evaluation was previously done by looking at the average of the area under the precision recall curve (AUPR) or the area under the receiver operating characteristic (AUROC) over the five data sets [37, 45]. Any method that provides a ranking of possible network edges could be evaluated in this framework.

We tested BETS and Enet against 22 other methods on the DREAM challenge [36, 37, 40, 49–51]. We ran SWING-RF, SWING-Lasso, CSId, Jump3, CLR, MRNET, and ARACNE in-house and found our results consistent with those reported in the literature. All 22 methods reported AUPR, but only 17 reported AUROC.

BETS ranked 7th out of 24 in AUPR with an average AUPR of 0.128 (Fig 2A and S1 Table) and 4th out of 19 in AUROC with an average AUROC of 0.688 (Fig 2B and S2 Table). BETS was the top performer of all VAR methods, and Enet was second best. All 24 methods outperformed random selection of edges, which achieved an average AUPR of 0.002 and average AUROC of 0.50 [49]. We also found that BETS and Enet had similar performance to the DBN methods in AUPR, and outperformed most of them in AUROC. Ranked by the top AUPR of each class of methods, the best performing class was DT, followed by GP, MI, VAR, DBN, and ODE [36, 40, 49]. The VAR method used in BETS produces edge signs (indicating excitatory or inhibitory causal effects) and effect sizes. While other methods based on GPs (e.g., CSId), MI (e.g., tl-CLR) or DTs (e.g., SWING-RF) had marginally better overall network inference, they do not provide insight into the causal relationships because they only output a positive measure of a causal interaction [32, 44, 51].

Next, we compared the speed of BETS and three other top-performing methods: SWING-RF, CSId and Jump3 (S3 Table). SWING-RF was the fastest at 0.11 hours, while BETS took 4.8 hours, CSId took 9.8 hours and Jump3 took 45 hours. Thus, while BETS had a lower AUPR compared to CSId and Jump3, it was substantially faster. BETS had both a lower AUPR and longer runtime than SWING-RF.

BETS improved upon Enet using stability selection. To quantify this improvement, we compared three other models: Elastic net with lag $L = 1$, ridge regression with lag $L = 2$, and lasso with lag $L = 2$ (S4 Table). In each case, the stability selection version outperformed the original version in average AUPR and AUROC. The improvement in average AUPR ranged between 0.016 and 0.03 (+20% to +31%), while the improvement in average AUROC ranged between 0.012 and 0.04 (+1.8% to +6.1%). Thus, our stability selection procedure leads to improved performance for multiple versions of VAR.

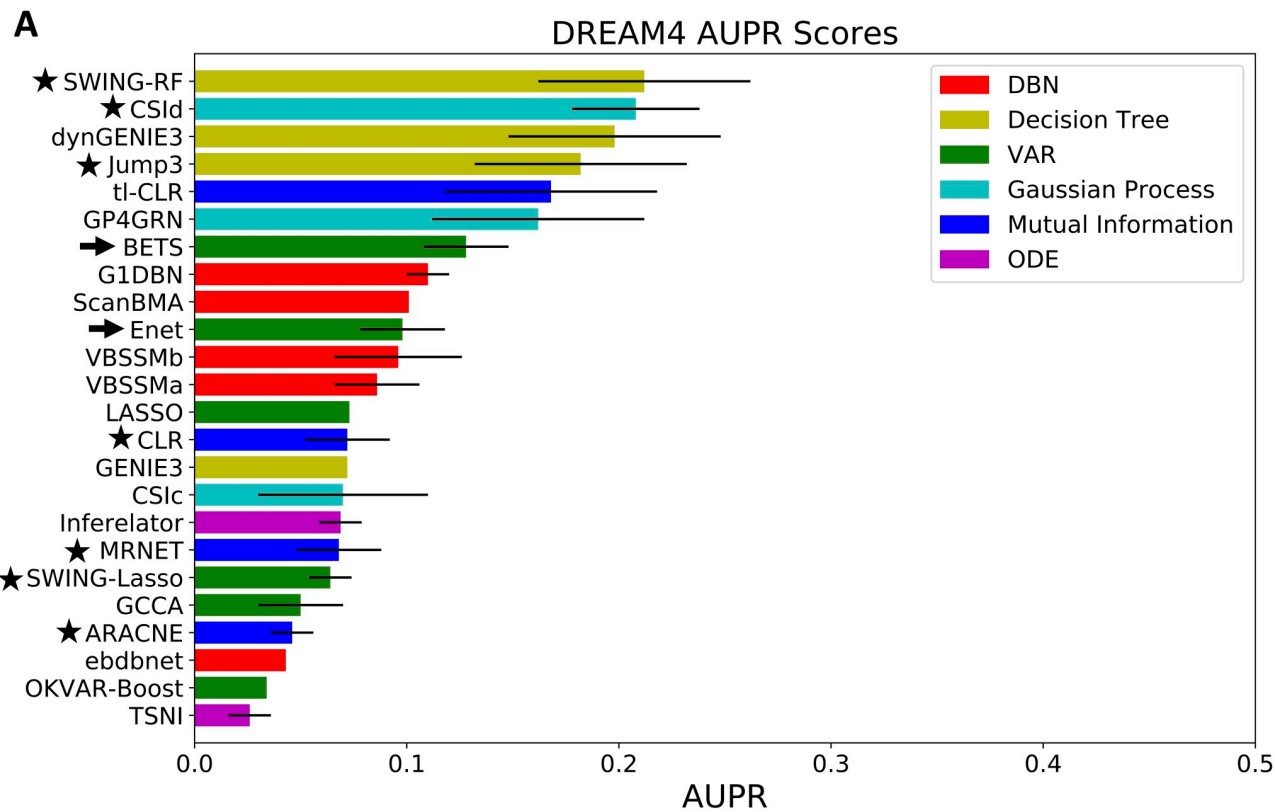

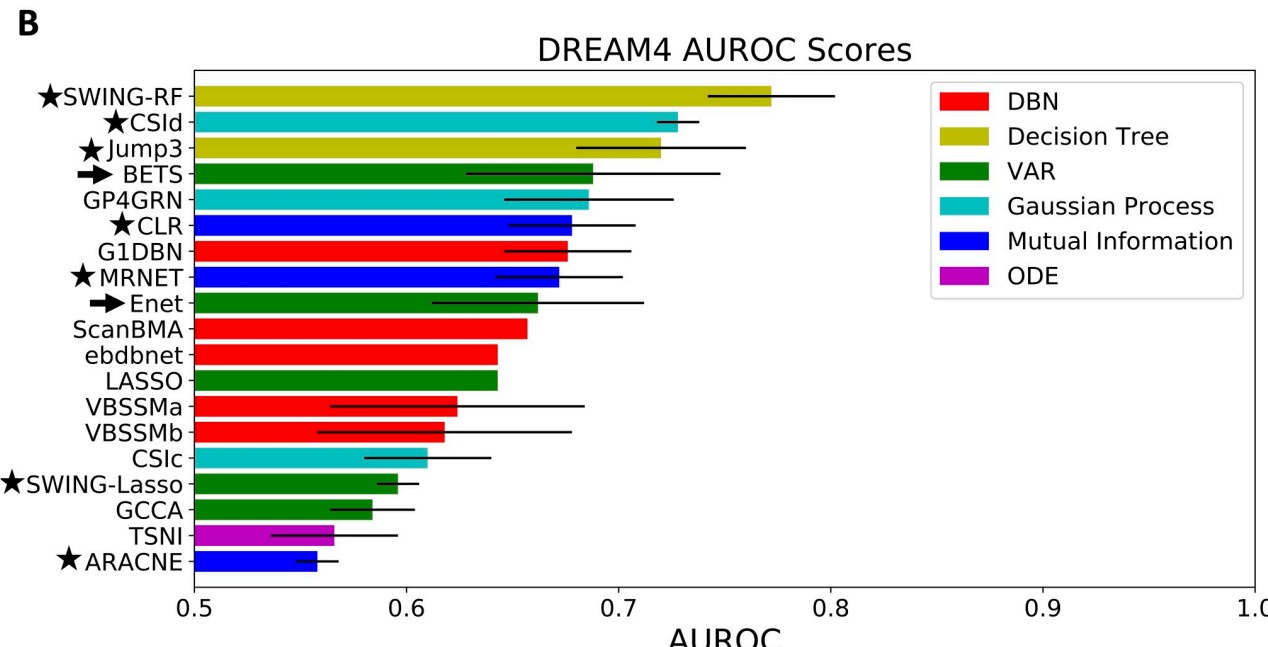

**Fig 2. Algorithm performance on the DREAM community benchmark. A) AUPR scores from 24 methods**, averaged across the five DREAM networks. **B) AUROC scores from 19 methods**, averaged across the five DREAM networks. Arrows indicate our methods. Stars indicate methods that we ran in-house; results were consistent with reported results. The bars reach one standard deviation from the average as calculated across the five DREAM networks; no bar indicates the standard deviation was not reported. See also S1–S5 Tables.

We also found that stability selection performance is robust to the number of bootstrap samples (S5 Table). Decreasing the number of bootstrap samples from 1000 to 100 led to minor decreases of −0.004 in AUPR and −0.008 in AUROC, within the standard deviation across the networks. It also resulted in a 10-fold decrease in memory usage and 3-fold decrease in run time, due to a constant-time hyperparameter search. If users face computational constraints, we recommend that they use 100 bootstrap samples for nearly equivalent performance.

Finally, we found that BETS' performance on DREAM is robust to the choice of lag. We ran BETS with lag $L = 1$ instead of lag $L = 2$ (S4 Table). BETS with lag $L = 1$ achieves an average AUPR of 0.14 (an increase of 0.012 from lag $L = 2$) and an average AUROC of 0.686 (a decrease of 0.002 from lag $L = 2$, within the standard deviation across networks). BETS with lag $L = 1$ still ranks 7th out of 24 in AUPR and 4th out of 19 in AUROC (tied with GP4GRN). Thus, BETS achieves consistently good performance on DREAM for both lags $L = 1$ and $L = 2$.

## 2.3 Application to gene transcription response to glucocorticoids

To infer the causal relationships in the GC response network, we analyzed RNA-seq data collected from human adenocarcinoma and lung model cell line A549, which consists of two data sets. In an *original exposure* data set, cells were exposed to the synthetic GC dexamethasone (dex) for 0, 0.5, 1, 2, 3, 4, 5, 6, 7, 8, 10, and 12 hours [6]. In an *unperturbed* data set, the cells were first exposed to dex for 12 hours, after which the media was replaced and dex removed, and then measurements were taken at the same intervals 0, 0.5, 1, 2, 3, 4, 5, 6, 7, 8, 10, and 12 hours. BETS was fit jointly over the two data sets. In total there were 7 technical replicates (4 from *original exposure* and 3 from *unperturbed*). A single VAR was fit on 70 samples: Each of the 7 replicates had 10 samples, because using a lag $L = 2$ VAR model turns 12 time points into 10 samples.

We applied BETS to the GC-mediated expression responses to infer a causal network (Fig 3A). Edges with selection frequency (i.e., frequency of appearance among bootstrap networks) at least 0.097 were declared significant (FDR ≤ 0.2; Fig 3B). The network contained 2, 768 nodes representing distinct genes and 31, 945 directed edges (0.4% of possible edges). Of these, 466 genes were causes (i.e., had an outward directed edge) and all 2, 768 genes were effects (i.e., had an incoming directed edge). In Granger causality, and dynamical systems more generally, a causal gene $g'$ is allowed to have incoming directed edges because $g'$ may be affected by the past value of another gene $g''$, and $g'$ may have a causal effect on the later value of gene $g$. The out-degree distribution was heavy-tailed and skewed right (Fig 3C), while the in-degree distribution was lighter-tailed and more symmetric (Fig 3D). The network's edge in-degree had a heavier left tail and lighter right tail than a normal distribution (Fig 3E). This suggests that causal genes are relatively rare (only 1/6th of network genes are causes) and a fifth of those only affect a single gene, whereas genes that are effects tend to have multiple causes. The network was inferred efficiently due to parallelization across genes, taking six days in real time and 292 days in CPU time to fit 5.5 million elastic net models.

To study the network with respect to the glucocorticoid system, we annotated specific genes as transcription factors (TFs), immune-related, or metabolism-related [52–55]. First, we inspected enrichment of each category among the causal genes (Fig 3F). At FDR ≤ 0.05, we found enrichment for TFs among causes; there were 226 causal TFs, representing 8.2% of the 2, 768 input genes. 62 of these TFs were causal, representing 13% of all causal genes (odds ratio (OR) = 2.0, Fisher's exact test (FET) adjusted $p \leq 2.9 \times 10^{-5}$). Similarly, we found an enrichment among immune-related genes as causes: of 109 immune genes, representing 3.9% of the input genes, 39 of these were causes, representing 8.4% of all causal

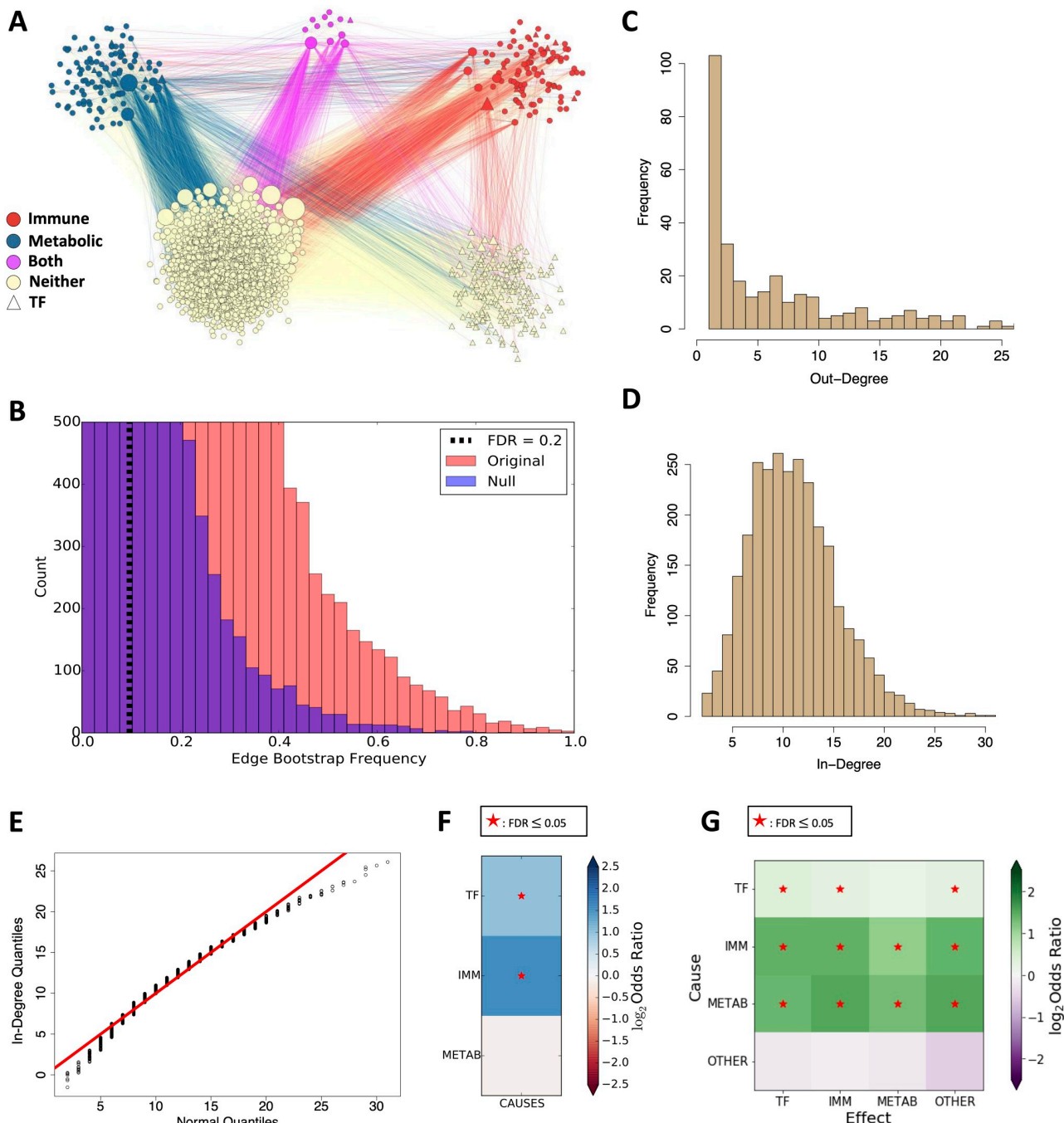

**Fig 3. Causal network inferred from glucocorticoid receptor data. A) Causal network clustered by gene type**. Edge color indicates the type of the causal gene: red edge indicates an immune causal edge, blue edge indicates a metabolic causal edge, purple edge (*both*) indicates an immune and metabolic causal edge, and tan edge indicates a neither immune nor metabolic other causal edge. **B) Significance thresholding for edges**, based on the null distribution of selection frequencies. **C) Out-degree distribution of network**. For clarity, several high out-degree values with low frequencies are not shown. **D) In-degree distribution of network**. **E) Quantile-quantile (Q-Q) plot of in-degree distribution against normal quantiles**. The in-degrees have a heavier left tail and lighter right tail than the normal distribution. **F) Enrichment of gene classes among network causal genes**, measured by odds ratio. **G) Enrichment of edge classes among network edges**, measured by odds ratio. See also S6 Table.

genes (OR = 2.9, FET adjusted $p \leq 2.5 \times 10^{-6}$). In contrast, there was no enrichment among metabolism-related genes: there were 120 metabolic genes, representing 4.3% of input genes; 19 of these metabolism genes were causes, representing 4.1% of all causes (OR = 0.93, FET adjusted $p \leq 0.66$). This suggests that our network is enriched for causal TFs and immune genes.

To study the interactions among gene classes inferred by our network, we quantified enrichment for edges between each of the four gene classes—immune, metabolic, TF, and other gene types (Fig 3G; S6 Table). We found enrichment of 11 of the 16 possible edge types (FDR $\leq 0.05$). The network was enriched for edges from i) causal TFs to immune genes and other genes; ii) causal immune genes to TFs, immune genes, metabolic genes, and other genes; and iii) causal metabolic genes to TFs, immune genes, metabolic genes, and other genes. This suggests that our network is enriched for a broad range of edge types.

The inferred relationships in BETS are conditionally Granger-causal, meaning that we find that gene $g'$ Granger-causes $g$ conditioning on the effects of other genes. Thus, an edge $g' \rightarrow g$ should be assessed based on $g$'s residuals after controlling for the effects of other genes and itself, instead of $g$'s raw values. Consider the edge $KRT6A \rightarrow NKAIN4$ (Fig 4): $NKAIN4$'s raw expression values suggest a negative relationship between $KRT6A$ and $NKAIN4$ (Fig 4A and 4B). However, after controlling for the effects of all covariates besides $KRT6A$, a positive relationship between $KRT6A$ and $NKAIN4$ appears (Fig 4C and 4D). Thus, conditionally Granger-causal relationships $g' \rightarrow g$ should assess $g$ only after controlling all other effects on $g$.

Our network identified known biological interactions between genes with immune, metabolic, and TF roles; we highlighted 16 gene pairs with experimentally validated interactions (Fig 5, S2 Fig, and S7 Table). Known interactions were found using the BIOGRID PPI database [56]. $SOCS1$ and $SOCS3$ bind $IRS2$ and promote its degradation, leading to reduced insulin signaling [57, 58]; furthermore, $SOCS1$ represses $IL$-$4$-induced $IRS2$ singling [59]. $NR4A1$ heterodimerizes with $RXRA$ to activate it in order to promote gene expression under vitamin A signaling [60]; $NR4A1$ also inhibits $p300$-induced $RXRA$ acetylation [61]. Eleven of the 16 edges had the correct interaction direction; the five that were reversed are $TNFAIP3 \rightarrow IRAK2$, $SOCS3 \rightarrow HIVEP1$, $ATF3 \rightarrow MDM2$, $E2F1 \rightarrow CDH1$, and $FOS \rightarrow EGFR$. These results suggest that BETS infers biologically meaningful relationships, but transcriptional data, absent other assays on protein abundance and cellular dynamics, are often underpowered to resolve the direction of the edge.

## 2.4 Validation of inferred network on overexpression data

We asked whether our inferred network edges validated on overexpression versions of the same experimental system, in which each of ten TFs was separately overexpressed over the same 12 hours of observations. Specifically, we assessed the concordance between inferred network edges $g' \rightarrow g$ and their coefficient in the overexpression data set under a VAR model (Methods).

We first evaluated how well network edges replicated on individual overexpression data sets. We performed linear regression of a one-hot encoding of the original network's edge sign (i.e., positive versus no edge or negative; negative versus positive or no edge) as the predictor against the VAR model edge coefficients estimated from each of the overexpression time series as the response (Fig 6A and 6B, Methods). Of the ten data sets, 9 showed enriched positive effect sizes among positive edges at FDR $\leq 0.2$ ($CEBPB$, $CEBPD$, $FOSL2$, $FOXO1$, $FOXO3$, $KLF6$, $KLF9$, $KLF15$, $OCT4$; Fig 6A). Three data sets showed enriched negative effect sizes among negative edges ($OCT4$, $TFCP2L1$, $CEBPD$) and four showed enriched positive effect sizes among negative edges ($CEBPB$, $FOSL2$, $KLF9$, $KLF15$; Fig 6B).

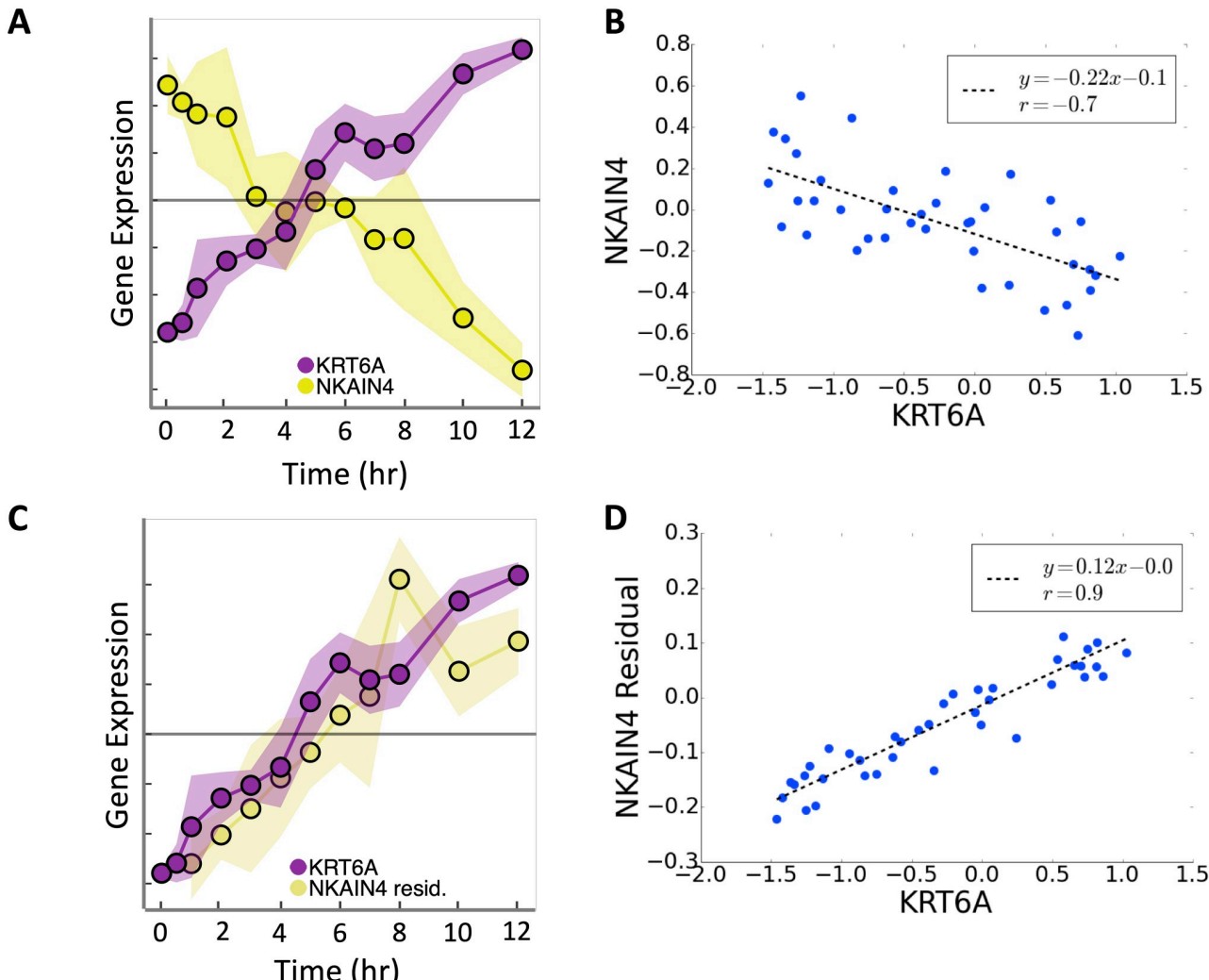

**Fig 4. Conditional Granger causality reveals opposite sign of relationship *KRT6A → NKAIN4*. A) Time series** and **B) scatter plot of expression values from *KRT6A* and *NKAIN4*. C) Time series** and **D) scatter plot of expression values from *KRT6A* and residual expression values from *NKAIN4*** after controlling for the effects of other covariates in *NKAIN4*. Each y-axis tick in A and C indicates 0.1 unit-variance standardized ln(TPM), where TPM is Transcripts Per kilobase Million. The grey line marks zero-centered expression. B and D axes are in units of ln(TPM).

Taken together, the positive edges inferred by BETS validate on the overexpression data, but the negative edges do not, indicating repressive effects may have inconsistent signs or feedback loops.

Next, we checked whether the 123 inferred causal edges from the TF *TFCP2L1* validated in the *TFCP2L1* overexpression data set (there were only about 10 causal edges from each of the other 9 TFs). We regressed the original network's edge sign (+ 1 for positive edges, 0 for no edge, and −1 for negative edge) as the predictor against the overexpression VAR model edge coefficients as the response (Fig 6C). We found a positive relationship between the edge sign and overexpression coefficient (slope 0.17, two-sided t-test $p \leq 5 \times 10^{-5}$). This shows that causal edges from *TFCP2L1* are enriched for matched effect directions in the *TFCP2L1* overexpression data.

This validation may be limited by a misspecification of the linear regression model. As a supplementary analysis, we fit nonstationary GPs to gene trajectories using *nsgp*; genes that

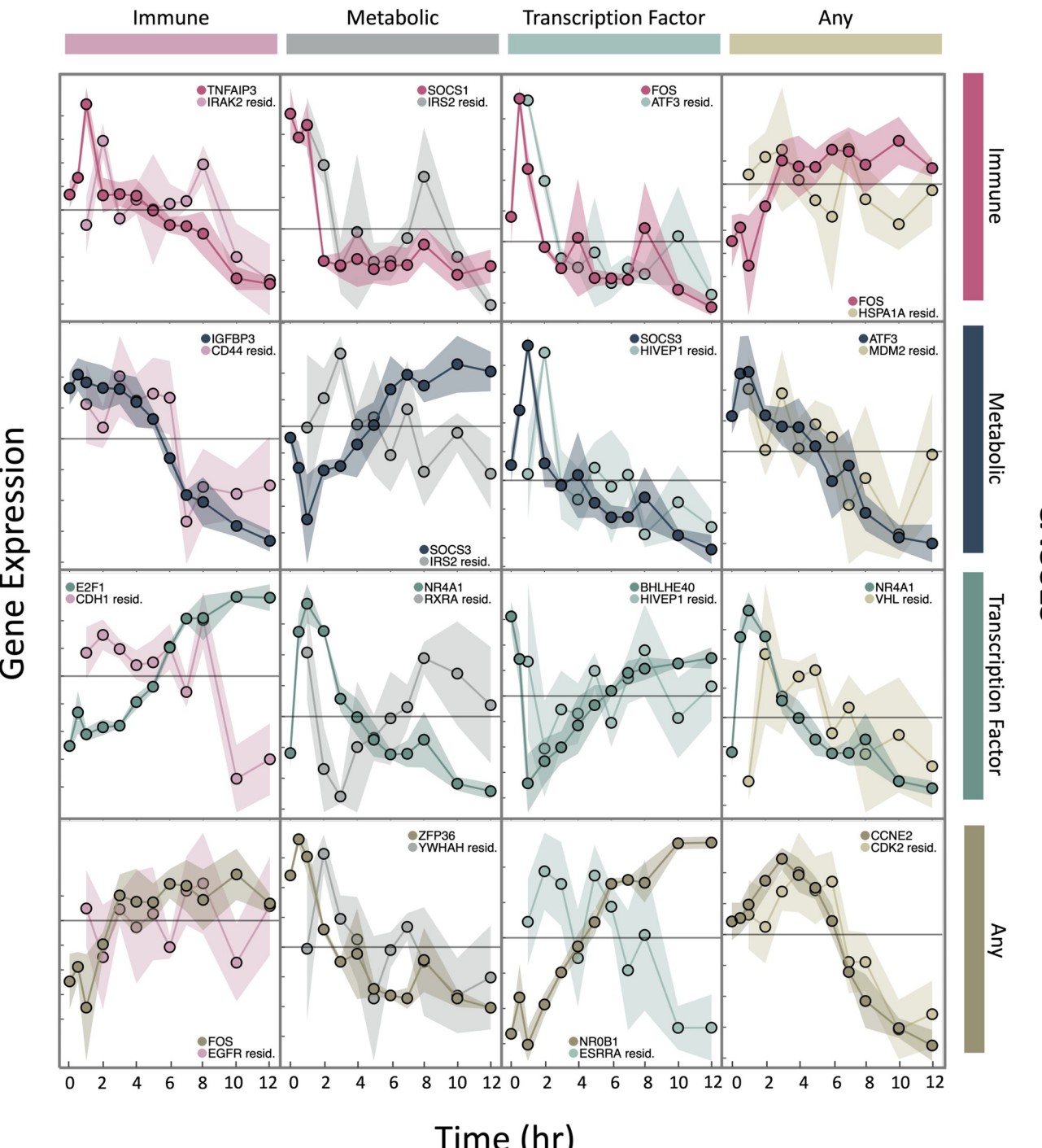

**Fig 5. Time-series profiles of experimentally validated causal interactions across gene classes.** For each gene pair, their profiles were from either the *original exposure* data set or the *unperturbed* data set. The effects of all covariates beside the causal gene were controlled in the effect gene values to show the conditional Granger-causal relationship. Colors encode gene classes: pink shows immune genes, dark blue/gray shows metabolic genes, teal shows TFs, and brown/tan shows other genes. Darker colors show causal genes and lighter colors show effect genes. The grey line marks zero-centered expression. Each y-axis tick indicates 0.1 unit-variance standardized ln(TPM). See also S7 Table and S2 Fig.

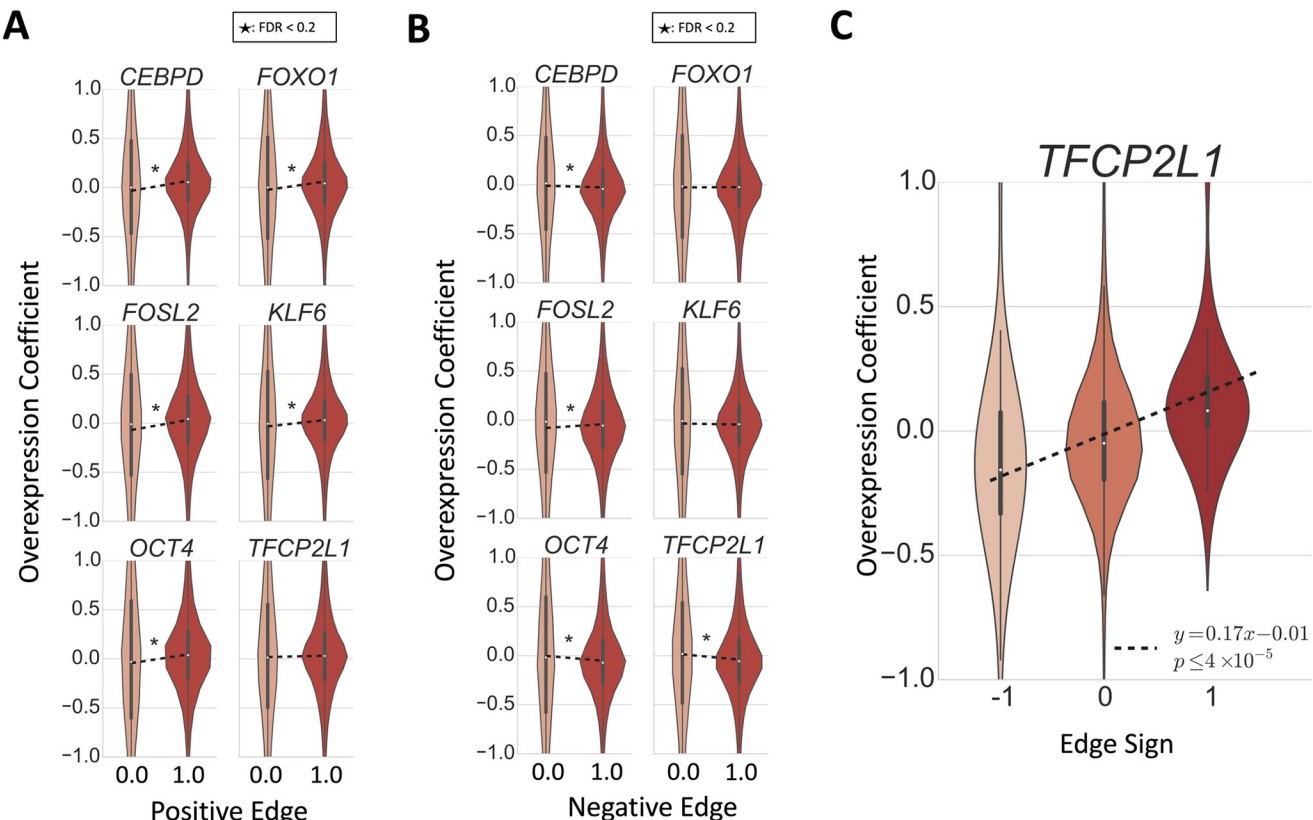

**Fig 6. Validation of inferred network on overexpression data. A-B) Regression of one-hot encoding of positive (negative for B) edges as the predictor against the VAR model edge coefficient from the overexpression data as the response**. A 1 indicates that an edge had a positive (in A) or negative (in B) coefficient in the original inferred network (FDR ≤ 0.2). **C) For the 123 causal edges from *TFCP2L1*, regression of edge sign as the predictor against the VAR model edge coefficient from *TFCP2L1* overexpression data as the response**.

were differentially expressed in the TF overexpression data compared to the original exposure data were listed as potential targets of that TF (Supplemental Information). We found that BETS weakly predicts potential targets of 5 of the 10 over-expression TFs. This approach did not show substantial improvement above random prediction of potential targets on these data (FDR ≤ 0.1).

## 2.5 Validation of network edges through lung trans-eQTLs

We next validated our network edges on an expression quantitative trait-loci (eQTL) study. A single nucleotide polymorphism (SNP) $S$ is an eQTL for a gene $g'$ if it is associated with $g'$'s expression level within a population. Given a true causal edge $g' \rightarrow g$, if a SNP $S$ is a local (cis-) eQTL for $g'$, $S$ might also be a distal (trans-) eQTL for $g$ [62]. We used gene expression levels in primary lung tissue ($n$ = 278) from the Genotype Tissue Expression (GTEx) project v6p [14]. We observed an enrichment of low trans-eQTL association p-values from the directed network compared to shuffling the variant labels (Fig 7A and 7B). This suggests our network captures more valid causal effects than expected by chance.

We next inspected specific associations and their corresponding edges. We found 340 trans-eQTL pairs in lung samples corresponding to 130 network edges (q-value FDR ≤ 0.2). There are more trans-eQTLs than edges because there are multiple cis-eQTLs for some causal genes

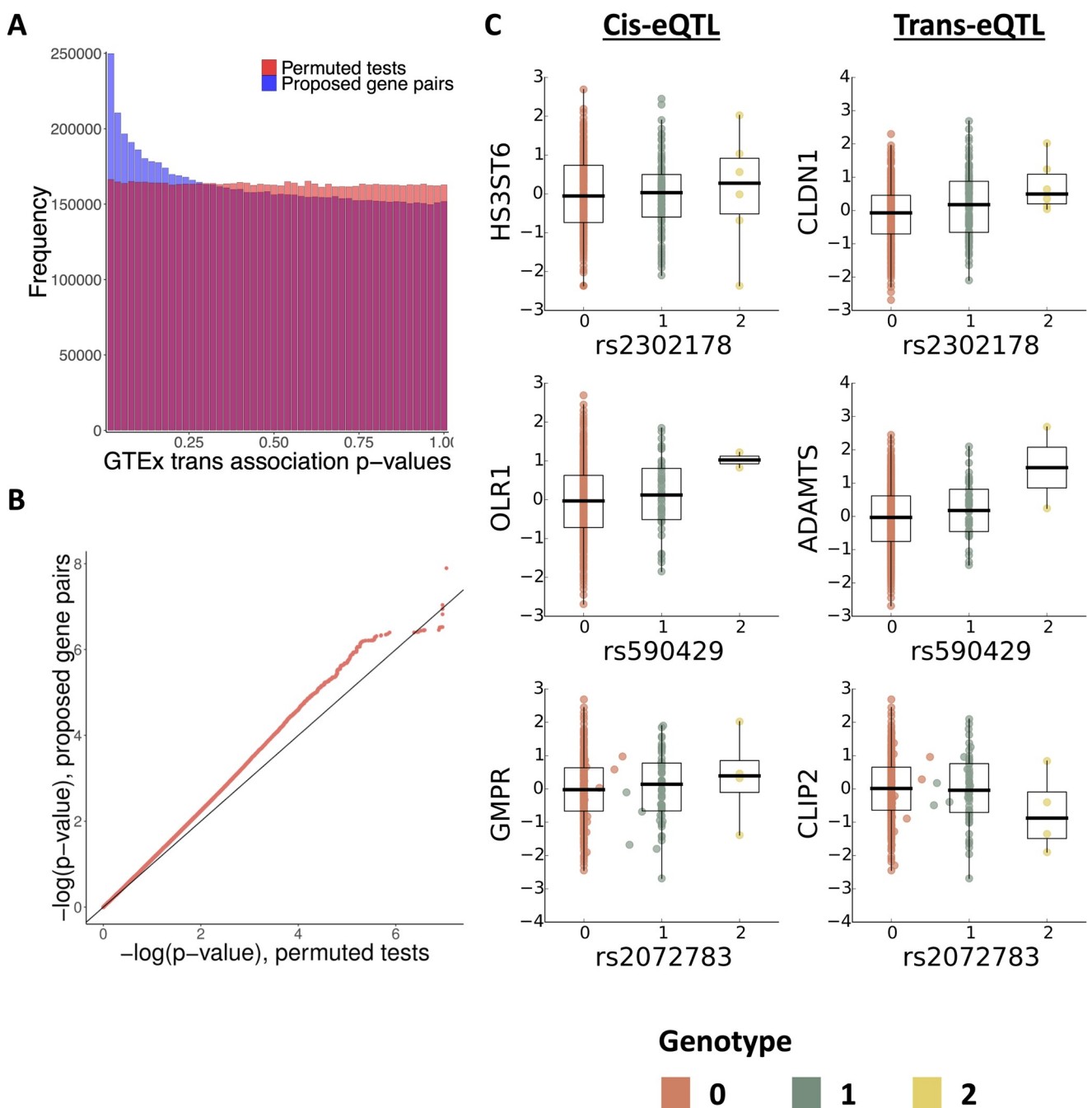

**Fig 7. Network edge validation using known cis- elements from GTEx v6 lung cis-eQTLs. A) Enrichment of trans associations in primary lung tissue among p-values from edges inferred by BETS** compared to p-values from permutations. **B) Quantile-quantile plot of validated edges** shows signal enrichment in lung samples when compared to signals from four other tissues in the GTEx v6 study. **C) SNPs associated with inferred gene pairs.** Genotype-phenotype plots corresponding to the cis-effect (left column), correlation in the GTEx v6 data between cause (y-axis) and effect (x-axis) gene pairs (right column).

$g'$. The 340 trans-eQTLs greatly improved upon the 2 identified in primary lung tissue in the GTEx v6 trans-eQTL study [63], demonstrating the utility of transcriptional time series for prioritizing promising associations. The top trans-associations were rs2302178-*CLDN1* (q-value FDR $\leq 0.095$; extended from the cis-association rs2302178-*HS3ST6*), rs590429-*ADAMTS*

(q-value FDR $\leq$ 0.11; extended from the cis-association rs2302178-*OLR1*), and rs2072783-*CLIP2* (q-value FDR $\leq$ 0.11; extended from the cis-association rs2302178-*GMPR*; Fig 7C).

We searched for validated associations between immune-related genes, metabolic-related genes, and TFs. One association was *OLR1* $\rightarrow$ *ITGAV*, where the known association between SNP rs4329754 and *OLR1* extends to an association between the same genetic variant and effect gene *ITGAV* (q-value FDR $\leq$0.13) [14]. *OLR1* plays key roles in immunity and metabolism [64, 65]. It is associated with metabolic syndrome [66] and atherosclerosis [66], and modulates inflammatory and humoral immune responses [67, 68]. Meanwhile, *ITGAV* plays a key role in the motility of $CD4^+$ T cells during inflammation [69].

Another association was between the TF *SNAI2* and gene *PTPN6*, where we find that the known association between genetic variant rs56800165 and *SNAI2* extends to an association between the same SNP rs56800165 and effect gene *PTPN* (q-value FDR $\leq$0.17) [14]. *SNAI2* is a direct target of the glucocorticoid receptor *GR* that regulates cell migration in breast cancer [70], while *PTPN6* is involved in glucose homeostasis via negative regulation of insulin signalling [71]. *PTPN6* is also associated with inflammatory phenotypes in multiple diseases [72, 73]. Finally, both *SNAI2* and *PTPN6* are involved in the cell-cell adherens junctions pathway, as *SNAI2* represses transcription of cadherin, while *PTPN6* positively regulates the cadherin-catenin complex [74]. Thus, for several eQTL-validated edges for gene pairs, we find that the genes are involved in related biological processes, but further experimentation is required to confirm direct interactions.

As A549 cells are models for lung tissue [75], we quantified enrichment of validated edges in lung compared to enrichment in four non-lung tissues with similar sample sizes: subcutaneous adipose ($n$ = 298), transformed fibroblasts ($n$ = 272), tibial artery ($n$ = 285), and thyroid ($n$ = 278). We validated 341 unique network edges across the five tissues (FDR $\leq$ 0.2). 130 edges validated for lung, 4 for subcutaneous adipose, 125 for transformed fibroblasts, 3 for tibial artery, and 82 for thyroid tissues. More network edges validated in primary lung than in non-lung tissues, suggesting that A549 cells most closely match lung samples among GTEx tissues; this is consistent with their tissue of tumor origin.

## 3 Discussion and conclusion

We described an approach, BETS, to build directed networks using short time-series observations of high-dimensional transcription data. BETS combined ideas from elastic net regression, graphical lasso, stability selection, and VAR models to infer Granger-causal relationships in high-dimensional transcription time-series data. Our method achieved competitive performance on the DREAM4 100-Gene Network Inference Challenge, ranking 7th out of 24 methods in AUPR and 4th out of 19 methods in AUROC; it was also faster than several methods with similar or better performance and infers effect size and sign, unlike the other top performing methods. Stability selection resulted in consistent improvement to VAR models across different hyperparameter settings.

Next, we applied BETS to time-series RNA-seq data from human A549 cells exposed to glucocorticoids and identified a directed network of 31, 945 edges (FDR $\leq$0.2), capturing the causal relationships among genes after exposure to GCs. Despite the intervention of GCs in this cell line, we expect that many of the causal relationships estimated using these data will exist across cellular environments in lung cells. In our estimated causal network, we found enrichment of immune genes and TFs among causal genes. We also found enrichment of 11 specific types of causal edges from TFs, immune genes, and metabolic genes. We validated our network first in ten overexpression data sets. Edges that were positive in the original network had an enrichment of positive VAR effect sizes in the overexpression data. However, edges in

the original network did not predict differential expression of genes in the overexpression data, as called by *nsgp*. Validating network edges by searching for trans-eQTLs in GTEx primary lung tissue samples, we found an enrichment of associations with genetic variants across network edges. Finally, we discovered 340 trans-eQTLs, a dramatic improvement from the GTEx v6 trans-eQTL study [63].

While BETS has demonstrated effective inference of causal relationships, there are interesting future directions to explore. All methods that infer networks from transcriptional time series face several difficulties: i) Transcript levels are sometimes an imperfect proxy for protein levels, especially when transcript dynamics are changing [15, 76]; ii) the scarcity of time point samples causes statistical challenges for inferring millions of possible causal interactions between genes, let alone non-additive interactions among causes [3, 77, 78]; iii) transcription data do not capture the complete regulatory context including chromatin structure and epigenetic regulations [15]; iv) transcription relationships are often nonstationary: the relationship may change over time due to responses from the environment [4, 5]; and v) inferred networks are often sensitive to the choice of preprocessing and parameter choices [79]. Single-cell data also implicitly include transcription time-series information when pseudotime is inferred, making ideas from Granger causality exceptionally relevant. However, recent preprints suggest some limits to pseudotime's potential for network inference [80, 81]. Finally, experimental followup is key to establishing causality; BETS only generates promising, interpretable hypotheses. Indeed, by discovering hundreds more trans-eQTLs than the GTEx study (a 170-fold increase) [63], BETS demonstrates its potential to prioritize biologically meaningful associations.

# 4 Methods

## 4.1 Method details

**Bootstrap Elastic net regression from Time Series (BETS).** Bootstrap Elastic net regression from Time Series (BETS) is a vector-autoregressive approach to causal inference from gene expression time-series data. It is based on the principle of Granger causality [31]: a gene $g'$ Granger-causes another gene $g$ if previous information from gene $g'$ improves our current predictions of gene $g$, beyond using previous information from $g$ and from other genes.

BETS first preprocesses the data. BETS fits an elastic net vector autoregression model to handle the high dimensionality of the time series, inferring a network (Fig 1A). It infers one network for each of 1000 bootstrapped samples of the original data set and computes each edge's selection frequency, or its frequency of appearance among the bootstrapped networks (Fig 1B) [3]. Finally, BETS includes an edge in the network using the selection frequencies (Fig 1B). Our baseline comparison, Enet, only preprocesses the data and fits an elastic net vector autoregression model from the original data (Fig 1A; Section 2.2).

**Preprocessing temporal time-series data.** For a **gene temporal profile** (i.e., one gene's expression values across time for a single replicate), we used zero-mean unstandardized normalization, which centers each gene temporal profile to have mean zero across time. Because gene temporal profile ranges from staying almost constant to having drastic fluctuations, BETS uses this approach because a unit-variance normalization would over-represent the weak causal effects of genes with lower variability.

**Vector autoregression model.** Let $G$ be the set of all genes in the data, let $p = |G|$ be the number of genes, and let $g$ be a gene. Let $\neg g$ be $G$ with $g$ removed. Let there be $T$ time points total, and let $t \in \{1, 2, \ldots, T\}$ be a single time point. Let there be $R$ replicates of the gene expression time series.

Let $X_{t,r}^g$ be the expression of gene $g$ at time $t$ for replicate $r$. Let $X_t^g = [X_{t,1}^g, X_{t,2}^g, \ldots, X_{t,R}^g]^T$ be the $R \times 1$ vector of gene expression levels of gene $g$ across $R$ replicates at time $t$. The rest of the paper does not mention replicates for simplicity, but here we discuss replicates for completeness.

Let $g'$ be the gene we are testing to be causal for gene $g$ and let $\ell$ refer to the time lag of the causal edge $g' \rightarrow g$. Let $L$ be the maximum lag. In BETS, $L = 2$ is the default.

We model each gene $g$ as

$$X_t^g = \sum_{\ell=1}^{L} \alpha_\ell^g X_{t-\ell}^g + \sum_{\ell=1}^{L} \sum_{g' \in \neg g} \beta_\ell^{g' , g} X_{t-\ell}^{g'} + \epsilon_t, \tag{2}$$

where $\epsilon_t \sim \mathcal{N}(0, 1)$. In other words, the expression of each gene $g$ is modeled as a linear function of its and other genes' $L$ previous expression values, under independent Gaussian noise. $\alpha_\ell^g$ represents the (scalar) effect size of gene $g$'s $\ell$th previous value, $X_{t-\ell}^g$, on its current value, $X_t^g$. $\beta_\ell^{g' , g}$ represents the (scalar) effect size of the $\ell$th previous value of gene $g' \neq g$, $X_{t-\ell}^{g'}$, on gene $g$'s current value, $X_t^g$. Eq 2 requires that $t > \ell$ for the $\ell$th previous value, $X_{t-\ell}^g$, to exist.

To demonstrate how our model is fit in practice, we reformulate Eq 2 using matrix notation. Each row represents one time point for one replicate. There are $T - L$ time points with $t > L$ and $R$ replicates, so there are $R(T - L)$ samples, or rows, in total. Let $N = R(T - L)$.

Define $\mathbf{X}_t^g$, an $N \times 1$ vector, as:

$$\mathbf{X}_t^g = \begin{bmatrix} X_{L+1,1}^g \\ \vdots \\ X_{L+1,R}^g \\ X_{L+2,1}^g \\ \vdots \\ X_{L+2,R}^g \\ \vdots \\ \vdots \\ X_{T,R}^g \end{bmatrix}. \tag{3}$$

We can similarly write $\mathbf{X}_{t-\ell}^g$, which is $\mathbf{X}_t^g$ with each entry replaced by its $\ell$th previous value. Define $\mathbb{X}_{t-\ell}^g$, a $N \times L$ matrix consisting of the first $L$ previous vectors $\mathbf{X}_{t-\ell}^g$, i.e., for $\ell$ ranging in $\{1, \ldots, L\}$.

$$\mathbb{X}_{t-\ell}^g = [\mathbf{X}_{t-1}^g \cdots \mathbf{X}_{t-L}^g]. \tag{4}$$

Let $\alpha_\ell^g$ be a $L \times 1$ vector of the $L$ lagged coefficients.

$$\alpha_\ell^g = \begin{bmatrix} \alpha_1^g \\ \vdots \\ \alpha_L^g \end{bmatrix}. \tag{5}$$

Next, let us formulate Eq 2 involving the genes $g'$ in matrix notation. Let $\mathbb{X}_{t-\ell}^{\neg g}$ be a $N \times L(|G| - 1)$ predictor matrix of the vectors $\mathbf{X}_{t-\ell}^{g'}$, for $g' \neq g$ and $\ell \in \{1, \ldots L\}$. Note the number of

columns is $L(|G| - 1)$, because there are $L$ previous time points $\ell \in \{1, \ldots, L\}$, and for each $\ell$, there are $|G| - 1$ genes $g' \neq g$, giving $|G| - 1$ vectors: $\mathbf{X}_{t-\ell}^{g'_1}, \ldots, \mathbf{X}_{t-\ell}^{g'_{|G|-1}}$.

$$\mathbb{X}_{t-\ell}^{\neg g} = \left[ \mathbf{X}_{t-1}^{g'_1} \cdots \mathbf{X}_{t-1}^{g'_{|G|-1}} \mathbf{X}_{t-2}^{g'_1} \cdots \mathbf{X}_{t-2}^{g'_{|G|-1}} \cdots \cdots \mathbf{X}_{t-L}^{g'_{|G|-1}} \right]. \tag{6}$$

Let $\beta_\ell^{\cdot g}$ be a $L(|G| - 1) \times 1$ vector of the causal coefficients $\beta_\ell^{g' \cdot g}$ where $g' \neq g$.

$$\beta_\ell^{\cdot g} = \begin{bmatrix} \beta_1^{g'_1 \cdot g} \\ \vdots \\ \beta_1^{g'_{|G|-1} \cdot g} \\ \beta_2^{g'_1 \cdot g} \\ \vdots \\ \beta_2^{g'_{|G|-1} \cdot g} \\ \vdots \\ \vdots \\ \beta_L^{g'_{|G|-1} \cdot g} \end{bmatrix}. \tag{7}$$

We then fit the model:

$$\mathbf{X}_t^g = \mathbb{X}_{t-\ell}^g \alpha_\ell^g + \mathbb{X}_{t-\ell}^{\neg g} \beta_\ell^{\cdot g} + \epsilon_t, \tag{8}$$

where $\epsilon_t$ is a $N \times 1$ vector with each element $\epsilon_{t,n} \sim \mathcal{N}(0, 1)$. In its most compact form, we can write

$$\mathbb{X}_{t-\ell}^G = [\mathbb{X}_{t-\ell}^g \mathbb{X}_{t-\ell}^{\neg g}], \quad \bar{\beta}^g = \begin{bmatrix} \alpha_\ell^g \\ \beta_\ell^{\cdot g} \end{bmatrix}. \tag{9}$$

Note that $\mathbb{X}_{t-\ell}^G$ is a $N \times L|G|$ matrix and $\bar{\beta}^g$ is a $L|G| \times 1$ vector. Thus, the matrix formulation of Eq 2 is:

$$\mathbf{X}_t^g = \mathbb{X}_{t-\ell}^G \bar{\beta}^g + \epsilon_t. \tag{10}$$

**Elastic net penalty.** Because of the large number of predictors as compared to the small number of samples, we use the elastic net penalty, which is a generalization of both ridge and lasso penalties. The elastic net fits the following objective:

$$\hat{\beta}_{\text{ELASTIC NET}}^g = \underset{\bar{\beta}^g \in \mathbb{R}^{L|G|}}{\arg\min} \|\mathbf{X}_t^g - \mathbb{X}_{t-\ell}^G \bar{\beta}^g\|_2^2 + \lambda(a\|\bar{\beta}^g\|_1 + (1 - a)\|\bar{\beta}^g\|_2^2). \tag{11}$$

Here $\| \cdot \|_1$ represents the $\ell_1$-norm and $\| \cdot \|_2$ represents the $\ell_2$-norm.

For the elastic net, we used the following ranges of hyperparameter values: $\lambda \in \{10^{-4}, 10^{-3}, \ldots, 1\}$, $a \in \{0.1, 0.3, \ldots, 0.9\}$. For lasso, we used $\lambda \in \{10^{-5}, \ldots, 1\}$. For ridge, when we used $\{10^{-5}, \ldots, 1\}$, we found that the optimal value selected in some cases was the maximum value of $\lambda = 1$. We thus expanded the range to $\{10^{-5}, \ldots, 10^6\}$ to ensure that we were not missing better hyperparameters at larger values. At this point, the optimal $\lambda$ was found to be 100.

**Hyperparameter tuning.** Hyperparameters were selected using leave-one-out cross-validation (LOOCV). The hyperparameter (or pair of hyperparameters, for elastic net) that minimizes the mean-squared error on the held-out observations is selected. More specifically, we first fix a hyperparameter $(\lambda, a)$. Then, for a given gene $g$ and row index $i$, extract the $i$th row of $\mathbf{X}_t^g$ and $\mathbb{X}_{t-\ell}^G$. We refer to this extracted validation set as $(\mathbf{X}_t^g)_i$ (target) and $(\mathbb{X}_{t-\ell}^G)_i$ (predictors). The remaining data is the training set, $(\mathbf{X}_t^g)_{-i}$ (target) and $(\mathbb{X}_{t-\ell}^G)_{-i}$ (predictors).

First, let $\hat{\beta}_{(\lambda,a),i}^g$ be the $\hat{\beta}_{\text{ELASTIC NET}}^g$ that is fit from the training set.

$$\hat{\beta}_{(\lambda,a),i}^g = \underset{\bar{\beta}^g \in \mathbb{R}^{L|G|}}{\arg\min} \|(\mathbf{X}_t^g)_{-i} - (\mathbb{X}_{t-\ell}^G)_{-i}\bar{\beta}^g\|_2^2 + \lambda(a\|\bar{\beta}^g\|_1 + (1-a)\|\bar{\beta}^g\|_2^2). \tag{12}$$

We then compute prediction error on the validation set, $\|(\mathbf{X}_t^g)_i - (\mathbb{X}_{t-\ell}^G)_i\hat{\beta}_{(\lambda,a),i}^g\|_2^2$. We repeat the fit $\hat{\beta}_{(\lambda,a),i}^g$ and error for every row index $i$ of $\mathbf{X}_t^g$ and for every gene $g$. The mean held-out cross-validation error for $(\lambda, a)$ is:

$$MSE(\lambda, a) = \sum_{g \in G} \sum_{i=1}^{N} \frac{1}{N} \|(\mathbf{X}_t^g)_i - (\mathbb{X}_{t-\ell}^G)_i\hat{\beta}_{(\lambda,a),i}^g\|_2^2. \tag{13}$$

The $(\lambda, a)$ that minimizes the error in Eq 13 is selected.

**Permuted coefficients.** We evaluate the significance of any given edge $g' \to g$ through permutation. In detail, we remove the time dependency between $g'$ and $g$ via permutations of individual gene temporal profiles over time.

We first generate a single permuted data set $\tilde{X}_t^g$. For each gene, we independently shuffle the temporal profile of each gene $g \in \{1, \ldots, |G|\}$ across time (Fig 1A). This is done separately for distinct replicates.

We wish to model the hypothesis of no causal relations from any gene $g' \in \neg g$, upon a given effect gene $g$. We use the unpermuted values of the effect gene $X_t^g$ and the permuted values of all other causal genes $g' \in \neg g$, as $\tilde{X}_t^{\neg g}$. The effect gene $g$ remains unpermuted, as we do not consider self-regulatory loops.

Permutation-based causal coefficients $\tilde{\beta}_\ell^{g',g}$ are then fit as

$$X_t^g = \sum_{\ell=1}^{L} \alpha_\ell^g X_{t-\ell}^g + \sum_{\ell=1}^{L} \sum_{g' \in \neg g} \tilde{\beta}_\ell^{g',g} \tilde{X}_{t-\ell}^{g'} + \epsilon_t. \tag{14}$$

We use these coefficients to perform FDR calibration.

**Edge FDR.** The result of the elastic net VAR model is a complete network whose edges are weighted according to the estimated regression coefficients.

For each lag $\ell \in \{1, \ldots, L\}$ and effect gene $g$, we control the edge FDR at $\leq 0.05$ by finding the threshold $T_\ell^g$ such that

$$\frac{\sum_{g' \in \neg g} \mathbb{1}\{|\tilde{\beta}_\ell^{g',g}| > T_\ell^g\}}{\sum_{g' \in \neg g} \mathbb{1}\{|\tilde{\beta}_\ell^{g',g}| > T_\ell^g\} + \sum_{g' \in \neg g} \mathbb{1}\{|\beta_\ell^{g',g}| > T_\ell^g\}} \leq 0.05. \tag{15}$$

For each gene pair $(g', g)$, $g' \in \neg g$, a directed edge $g' \to g$ exists if, for at least one of the lags $\ell \in \{1, \ldots, L\}$, $|\beta_\ell^{g',g}| > T_\ell^g$.

**Stability selection.** Stability selection is used to ensure the robustness of BETS to small sample size. Stability selection is a method for high-dimensional graph estimation that uses bootstrap samples [82]. While the authors prove finite sample control for the family-wise error rate (FWER), we are interested in controlling the false discovery rate (FDR).

This procedure draws $B = 1000$ bootstrap samples, where each sample consists of $N$ rows drawn with replacement from output $\mathbf{X}_t^g$ and input $\mathbb{X}_{t-\ell}^G$ (Eq 10). Let the $j$th bootstrap sample from the original data be $\mathbf{X}_t^{g,j}$ and $\mathbb{X}_{t-\ell}^{G,j}$. A set of $N$ row indices, $I^j$, are sampled with replacement from $[1, 2, \ldots, N]$. $\mathbf{X}_t^{g,j}$ and $\mathbb{X}_{t-\ell}^{G,j}$ are created by choosing the rows $I^j$ of $\mathbf{X}_t^g$ and $\mathbb{X}_{t-\ell}^G$. $\mathbf{X}_t^{g,j}$ is an $N \times 1$ vector and $\mathbb{X}_{t-\ell}^{G,j}$ is an $N \times L|G|$ matrix.

Now consider the permuted output and input, $\tilde{\mathbf{X}}_t^g$ and $\tilde{\mathbb{X}}_{t-\ell}^G$, constructed from $\tilde{X}_t^g$ (Eq 10). Let the $j$th bootstrap sample from the permuted data be $\tilde{\mathbf{X}}_t^{g,j}$ and $\tilde{\mathbb{X}}_{t-\ell}^{G,j}$. $\tilde{\mathbf{X}}_t^{g,j}$ and $\tilde{\mathbb{X}}_{t-\ell}^{G,j}$ are created by choosing the rows $I^j$ of $\tilde{\mathbf{X}}_t^g$ and $\tilde{\mathbb{X}}_{t-\ell}^G$.

Thus, the $j$th bootstrap sample for both the original and permuted data sets use the same row indices $I^j$.

For each of the 1000 bootstrap samples, we infer a network using the elastic net fit and edge FDR procedure described earlier. Each edge $g' \rightarrow g$'s selection frequency, $\pi_{g', g}$ (the frequency of $g' \rightarrow g$ among the 1000 bootstrap networks) is computed (Fig 1B).

**Stability FDR.** To determine the appropriate cutoff for the selection frequency of each edge ($\pi_{g', g}$), we generate a null distribution of selection frequencies using permutations. First, we generate a second permuted data set $\hat{X}_t^g$ in which we again independently shuffle the temporal profile of each gene $g \in \{1, \ldots, |G|\}$ across time. This is done separately for distinct replicates.

We run the stability selection procedure on $\hat{X}_t^g$ as if it were $X_t^g$, using the same set of row indices $I^j$ to generate the bootstrap samples, and using $\tilde{X}_t^g$ to generate the permuted coefficients.

After running for all $B = 1000$ bootstrap samples, we obtain the null selection frequency of each edge, $\hat{\pi}_{g'.g}$.

We control the stability FDR at 0.2 by finding the threshold $T_b$ such that

$$\frac{\sum_{g' \in \neg g} \mathbb{1}\{\hat{\pi}_{g'.g} > T_b\}}{\sum_{g' \in \neg g} \mathbb{1}\{\hat{\pi}_{g'.g} > T_b\} + \sum_{g' \in \neg g} \mathbb{1}\{\pi_{g'.g} > T_b\}} \leq 0.2. \tag{16}$$

Because the maximum lag is 2, each edge $g' \rightarrow g$ has two possible lags and thus two selection frequencies. The lag with larger absolute value of average coefficient across the 1, 000 networks is considered in both the permuted and the real empirical distributions. So, if $|\beta_1^{g'.g}|$ exceeds $|\beta_2^{g'.g}|$, the lag is said to be 1 and the selection frequency $\pi_{g'.g}^1$ is used.

**Network inference performance metrics.** Refer to every network edge inferred by a method as a positive and every missing edge as a negative. Let *TP* be True Positives, *FP* be False Positives, *TN* be True Negatives, and *FN* be False Negatives. Let *TPR* be True Positive Rate, (i.e., recall), and *FPR* be False Positive Rate. Then, we have

$$TPR = \frac{TP}{TP + FN}$$

$$FPR = \frac{FP}{FP + TN}$$

$$Precision = \frac{TP}{TP + FP}$$

In the DREAM benchmark, each network inference method is evaluated by comparing the true network (i.e., the network used to generate the synthetic data) with the inferred network

at different thresholds for edge inclusion. The two main evaluation metrics are Area Under the Receiver Operating Characteristic curve (AUROC) and Area Under the Precision-Recall curve (AUPR). AUROC plots TPR on the *y* axis and FPR on the *x* axis. AUPR plots precision on the *y* axis and recall on the *x* axis. When the number of negatives greatly exceeds the number of positives, as with gene networks, which are typically sparse, AUPR is a more relevant metric [83].

## 4.2 Software

BETS is available for download on Github at https://github.com/lujonathanh/BETS. The software is licensed under the terms of the Apache License, version 2.0. The analysis code is available at https://zenodo.org/record/4009546#.X5XEh0JKg1g.

## 4.3 Data sets and processing

**DREAM Network Inference Challenge.**   There were five data sets in the DREAM4 Network Inference Challenge, each consisting of ten time series of 21 time points and 100 genes [45, 84]. For the first half of the time series, a "drug perturbation" was applied; this affected about 1/3 of genes. For the second half, the perturbation was removed and the system was allowed to relax back to the wild-type state.

**Glucocorticoid gene expression data.**   We analyzed RNA-sequencing data from a set of experiments developed to study glucocorticoid receptors (GRs) in the human adenocarcinoma and lung model cell line, A549 [6]. There was an *original exposure* data set of 4 replicates in which cells were stimulated by the glucocorticoid dexamethasone (dex), and gene expression was profiled at {0, 0.5, 1, 2, 3, 4, 5, 6, 7, 8, 10, 12} hours of dex stimulation. There was also an *unperturbed* data set of 3 replicates in which cells were exposed to dex for 12 hours, after which the conditioned media was replaced and dex removed. Gene expression was profiled at {0, 0.5, 1, 2, 3, 4, 5, 6, 7, 8, 10, 12} hours after dex removal. We integrated the *original exposure* and *unperturbed* data into a joint data set with 7 replicates. The *original exposure* data set is available at the Gene Expression Omnibus (GEO), with reference numbers listed for the rows that list "RNASeq" as the Assay under the column "Experiment_GEO_Series" in Supplementary Table 3 of [6]. The GEO accession numbers for time points {0, 0.5, 1, 2, 3, 4, 5, 6, 7, 8, 10, 12} of the *original exposure data* set are GSE91305, GSE91198, GSE91311, GSE91358, GSE91303, GSE91243, GSE91281, GSE91229, GSE91255, GSE91284, GSE91222, and GSE91212, respectively. The *unperturbed* data set is available at GEO accession number GSE144662.

We selected 2768 genes for analysis, which had average expression > 2 Transcripts Per kilobase Million (TPM) and were differentially expressed in the original exposure data. A gene was called differentially expressed if its expression at any time point differed from its expression at time 0, ascertained by running edgeR (FDR $\leq$ 0.05) [6]. We added *NR3C1*, which encodes the glucocorticoid receptor (GR). *NR3C1* was not found to be differentially expressed at FDR $\leq$ 0.05.

After genes were selected, gene expression TPM were log-normalized and corrected for surrogate variables using SVAseq [85]. Each gene's temporal profile was centered to have mean zero across time. In the *original exposure data*, all replicates besides replicate 1 had a measurement for each time point. Replicate 1 was missing time points 5 and 6 hrs, so we imputed these values using a linear interpolation from time points 4 and 7 hrs in the log-transformed, surrogate-corrected space.

**Overexpression transcriptional time-series data.**   There were ten *overexpression* data sets, in which each of the transcription factors *CEBPB*, *CEBPD*, *FOSL2*, *FOXO1*, *FOXO3*,

*KLF6*, *KLF9*, *KLF15*, *POU5F1*, and *TFCP2L1* was separately overexpressed across 12 hours of dex stimulation. Each overexpression data set had three replicates; gene expression was profiled after {0, 1, 4, 8, 12} hours of dex stimulation. The same 2768 genes were selected and the same normalization and SVAseq correction as earlier was performed. The *overexpression* data sets are available at the Gene Expression Omnibus at GEO accession number GSE144660.

## 4.4 Application of methods to the data

**DREAM benchmarking.** We ran the methods BETS, Enet, CSId [44], Jump3 [36], CLR [27], MRNET [28], ARACNE [29], SWING-RF [51], and SWING-Lasso [51] on the DREAM challenge. In BETS, inferred edges were ranked by their selection frequency for calculating AUPR and AUROC. In Enet, edges were ranked by the absolute value of their coefficient. The Python3 version of CSId was run after obtaining it from correspondence with Dr. Penfold.

Jump3 required setting the "systematic noise" and "observational noise" parameters. We used Dr. Huynh-Thu's settings on the DREAM challenge, with systematic noise at $1e - 4$ and observational noise at 0.01 times the value of the gene's expression. ARACNE, MRNET, and CLR were run using the minet R library. BETS, Enet, CSId, and Jump3 were run on a single node without parallelization. The node had 28 cores, 128 GB of memory, and 2.4 GHz processor speed. ARACNE, MRNET, and CLR were run on a 4 GB RAM, Intel Core i5 1.3 GHz laptop.

**Network analysis: Gene annotations.** We considered genes with three possible labels: immune system, metabolism, or transcription factor. Immune genes were labeled as such using two sources. The first source is the Gene Ontology (GO) annotation "Immune" (*GO:0002376*) [52]. We applied this label when the evidence codes were one of EXP, IDA, IGI, IMP, IPI, IC, TAS. The second source is the Gene Ontology Consortium's curated, ranked list of immune-related genes based on multiple databases and experimental evidence [54]. For the GO annotation, we selected all genes with score $\geq 7$. This resulted in 616 immune genes overall, and 109 immune genes in our list of 2768 genes.

Metabolic genes were called using two sources. The first source is the GO annotation "carbohydrate metabolic process" *GO:0005975* [52]. We applied this label when the evidence codes were EXP, IDA, IGI, IMP, IPI, IC, TAS. The second source is the Gene Set Enrichment Analysis (GSEA)-curated list of metabolic-related genes [53]. We searched only among those with experimental evidence: the Canonical, KEGG, BIOCARTA, and Reactome pathways. We used the following four search queries: "gluconeogenesis OR (glucose AND metabolism) OR glycolysis," "lipid AND metabolism," "Diabetes," "Obesity." This resulted in 544 metabolic genes overall, of which 120 were in our gene list. 65 genes were both immune and metabolic overall; 12 of these were in our gene list.

Transcription factors (TFs) were called using the Bioguo database of human TFs [55]. There were 1463 TFs overall, of which 226 were present in our gene list.

**Experimental interactions.** We created a list of experimentally validated interactions from the BIOGRID Homo sapiens Protein-Protein Interactions database [56]. Proteins were mapped to genes using BioMart from Ensembl 94 [86]. Among genes in our gene list, there were 17, 990 BIOGRID interactions.

**Validation on overexpression data.** The overexpression data had four time points with 1 to 4 hour time gaps, unlike the original 12 time points with 0.5 to 2 hour time gaps. On the overexpression data, we used a VAR model that regressed each effect gene's expression level on its previous expression level and the causal gene's previous expression level, assuming normal noise $\epsilon_t \sim \mathcal{N}(0, 1)$:

$$X_t^g = c^g X_{t-1}^g + d^{g',g} X_{t-1}^{g'} + \epsilon_t. \tag{17}$$

No regularization was included, and ordinary least squares was used to fit the equation. The expression $X_{t-1}^{g'}$ of a causal gene $g'$ is fit as a single predictor without the other expression. Lag 1, not 2, is used due to the larger time gaps.

**Validation on lung trans-eQTLs in GTEx v6.** Trans-eQTLs were discovered using the Genotype Tissue Expression (GTEx) v6 data [14, 63]. First, we mapped our genes from hg38 to hg19. For every edge $g' \rightarrow g$, we tested the set of genetic variants within 20 kilobases of $g'$ for trans-eQTL association with $g$ [87]. Specifically, we computed the p-value for linear association of each variant with the corresponding effect gene $g$ using MatrixEQTL [88]. A null distribution was generated by taking every edge $g' \rightarrow g$, permuting the effect gene $g$'s expression values, and repeating the linear association test. FDR over test statistics was calculated using q-value [89]. Because not every causal gene $g'$ had a cis-eQTL, only 26,839 edges (84% of the original 31,945 edges) were tested.

## 5 Supporting information

**S1 Fig. Overview of gene regulatory network inference methods.** Panels show each inference method applied to a cause gene $g'$ (blue, solid) and an effect gene $g$ (blue, dotted). A) Mutual information is computed between the cause and effect. B) The effect's expression is fit as an autoregression from the cause's past expression. C) The effect's expression is fit as a differential equation from the cause's current expression. D) The effect's expression is fit as a decision tree function of the cause's past expression. E) The space of dynamic causal networks is searched, with linear relationships between cause and effect. F) The space of dynamic causal networks is searched, with nonlinear relationships between cause and effect.
(PDF)

**S2 Fig. Causal gene expression and effect gene residuals from experimentally validated interactions.** On the y-axes are the effect gene residual expression values after subtracting the effects of all other covariates. Axes are in units of ln(TPM). Related to Fig 5 and S7 Table.
(PDF)

**S1 Table. DREAM4 100-gene network inference results, AUPR.** DBN is dynamic Bayesian network, DT is decision tree, GP is Gaussian process, MI is mutual information, ODE is ordinary differential equation, VAR is vector autoregression. The references that reported ebdbnet, ScanBMA, and LASSO did not provide AUPR values for individual networks. Algorithms that were run in-house were ARACNE, BETS, CLR, CSId, Enet, Jump3, MRNET, SWING-Lasso, and SWING-RF. Where reported literature values were available, they were consistent with these values. Values for CSIc, G1DBN, GCCA, GP4GRN, TSNI, VBSSMa and VBSSMb were taken from [49]. Values for ebdnet, LASSO and ScanBMA, were taken from [40]. Values for dynGENIE3, GENIE3, OKVAR-Boost and tl-CLR were taken from [37]. Values for Inferelator and Jump3 were taken from [36]. Related to Fig 2.
(DOCX)

**S2 Table. DREAM4 100-gene network inference results, AUROC.** DBN is dynamic Bayesian network, DT is decision tree, GP is Gaussian process, MI is mutual information, ODE is ordinary differential equation, VAR is vector autoregression. The references that reported ebdbnet, ScanBMA, and LASSO did not provide AUROC values for individual networks. Algorithms that were run in-house were ARACNE, BETS, CLR, CSId, Enet, Jump3, MRNET, SWING-Lasso, and SWING-RF. Values for CSIc, G1DBN, GCCA, GP4GRN, TSNI, VBSSMa and VBSSMb were taken from [49]. Values for ebdnet, LASSO, and ScanBMA, were taken from [40]. Related to Fig 2.
(DOCX)

**S3 Table. Results of in-house algorithms on DREAM4 100-gene network inference.** AUPR, AUROC, and Time indicate average AUPR, AUROC, and time over the five networks, respectively. BETS and Enet are in bold to indicate that they are our own developed methods, based on vector autoregression. SWING-RF [51] and Jump3 [36] are decision tree methods. CSId is a Gaussian process method [44]. CLR [27], MRNET [90], and ARACNE [29] are mutual information methods. SWING-Lasso is a vector autoregression method [51]. Related to Fig 2.
(DOCX)

**S4 Table. Improvement on DREAM4 100-gene network inference from bootstrap.** For each AUROC or AUPR column, the average is the listed value and the standard deviation is listed in parentheses. "Coefficient" denotes the result when ranking edges by their fitted coefficient, as in the original method. "Bootstrap" denotes the results when ranking edges by the frequency by which they appear in the bootstrap networks.
(DOCX)

**S5 Table. Dependency of BETS performance on bootstrap samples.** DREAM results reported for running BETS on both 100 and 1000 bootstrap samples. All values in the columns are averages and the parenthetical values as standard deviations across the 5 DREAM4 Networks. The 1000 samples row is bolded because 1000 samples are the default settings. These use zero-mean normalization, lag 2, and the elastic net penalty. Related to Fig 2.
(DOCX)

**S6 Table. Enrichment of edges between specific gene classes in inferred causal network.** A Fisher's Exact Test was performed, where the rows of the contingency table were whether or not an edge was of the edge type, and the columns were whether or not the edge was part of the inferred network. Related to Fig 3.
(DOCX)

**S7 Table. Gene pair information from Fig 5.** Shown Data Set indicates whether the gene temporal profiles in Fig 5 are taken from the *original exposure data* or *unperturbed data*. The edge type indicates the gene class of the causal and effect gene; for example, I → M indicates an edge from an Immune causal gene to a Metabolic effect gene. I = Immune; M = Metabolic; T = Transcription Factor; A = Any gene. Related to Fig 5.
(DOCX)

**S1 Text. Supplemental information.** Additional overview and analyses.
(DOCX)

## Acknowledgments

The authors would like to thank Gregory Darnell, Derek Aguiar, Ariel Gewirtz, Allison Chaney, Isabella Grabski, Cristina Anastase, and Genna Gliner for helpful discussion, feedback, and generosity in running cluster jobs; and Jian Peng for productive discussion and helpful comments.

The authors gratefully acknowledge that this work was performed using the Princeton Research Computing resources sponsored by the Princeton Institute for Computational Science and Engineering (PICSciE) at Princeton University.

## Author Contributions

**Conceptualization:** Bianca Dumitrascu, Barbara E. Engelhardt.

**Data curation:** Jonathan Lu, Ian C. McDowell, Alejandro Barrera, Linda K. Hong, Sarah M. Leichter, Timothy E. Reddy, Barbara E. Engelhardt.

**Formal analysis:** Jonathan Lu.

**Funding acquisition:** Timothy E. Reddy, Barbara E. Engelhardt.

**Investigation:** Jonathan Lu.

**Methodology:** Jonathan Lu, Bianca Dumitrascu, Barbara E. Engelhardt.

**Project administration:** Timothy E. Reddy, Barbara E. Engelhardt.

**Resources:** Linda K. Hong, Sarah M. Leichter.

**Software:** Jonathan Lu, Bianca Dumitrascu.

**Supervision:** Bianca Dumitrascu, Timothy E. Reddy, Barbara E. Engelhardt.

**Validation:** Jonathan Lu, Ian C. McDowell, Brian Jo, Barbara E. Engelhardt.

**Visualization:** Jonathan Lu, Barbara E. Engelhardt.

**Writing – original draft:** Jonathan Lu, Bianca Dumitrascu, Ian C. McDowell, Barbara E. Engelhardt.

**Writing – review & editing:** Jonathan Lu, Ian C. McDowell, Barbara E. Engelhardt.

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
