## [Decision Letter · Decision Letter 0]

16 Sep 2019

Dear Dr Engelhardt,

Thank you very much for submitting your manuscript 'Causal network inference from gene transcription time series response to glucocorticoids' for review by PLOS Computational Biology. Your manuscript has been fully evaluated by the PLOS Computational Biology editorial team and in this case also by independent peer reviewers. The reviewers appreciated the attention to an important problem, but raised some substantial critiques of the manuscript as it currently stands. While your manuscript cannot be accepted in its present form, we are willing to consider a revised version in which the issues raised by the reviewers have been adequately addressed. We cannot, of course, promise publication at that time.

In your revision, note in particular the suggestions for method comparison made by Reviewer #1 to demonstrate the practical advantages of the novel features of BETS, as well as technical and model interpretation questions from both reviewers. Please also clarify if a new RNA-seq data set has been generated for this study and if it has been deposited in a public repository.

Sincerely,

Christina S. Leslie

Associate Editor

PLOS Computational Biology

Thomas Lengauer

Methods Editor

PLOS Computational Biology

[LINK]

Reviewer's Responses to Questions

**Comments to the Authors:**

Reviewer #1: This manuscript presents the Bootstrap Elastic net regression from Time Series (BETS) algorithm, an network inference approach for time series gene expression data. The core question in time series-based network inference is how to use past information from regulators to detect their influence on the later expression levels of target genes. BETS adopts a Granger causality approach to address this question. Although Granger causality has been widely applied to time series gene expression data, there are multiple characteristics that differentiate BETS from prior work. The most notable is that BETS provides a false discovery rate (FDR) framework to rank and select network edges. Because it operates with temporal data, permuting the time points generates a convenient dataset in which the true regulator-target dependencies should be destroyed. Running BETS on the permuted data then creates a null distribution of regression coefficients or edge frequencies that can be used for FDR calculations. This FDR evaluation is combined with traditional bootstrapping and stability selection to make BETS robust to small sample sizes and false positive associations that are problematic in short time series. In addition, BETS incorporates an elastic net regression penalty and automated hyperparameter tuning, which are not implemented in many prior Granger causality methods.

In general, the presentation (figures, writing, review of related work, etc.), methodological choices, and analyses are superb. The synthesis of different types of time series network inference techniques in Figure S1 is an excellent high-level overview of the field. The evaluation on the simulated data DREAM4 challenge is valuable, demonstrating that BETS performs reasonably well on this popular benchmarking dataset and putting BETS in the context of many other types of network inference tools. There is a sizeable AUPR performance gap between BETS and the top method. However, this is not a concern because there are other advantages of BETS besides its AUPR on a simulated data set (e.g. parallelism, speed, and statistical rigor). Furthermore, other work in the gene network inference field has shown that AUPR on simulated data does not reflect the challenges of network inference in real mammalian systems, so maximizing AUPR on the DREAM4 data should not be the main objective of new methods.

Therefore, the glucocorticoid case study is more relevant and interesting. Because there are not complete gold standard networks available for evaluating condition-specific human regulatory networks, the BETS predictions are assessed with two independent datasets. Overexpression of 10 transcription factors demonstrates that there is general agreement between the gene expression changes induced by overexpression and the edge signs predicted by BETS. The BETS predictions are not perfect, as four transcription factors have positive effects enriched among negative predicted edges. In addition, trans-eQTL analysis of GTEx lung gene expression data reveals many new trans-eQTLs that are nominated by the BETS network edges. These demonstrate how BETS predictions can be used to gain biological insights.

One weakness with respect to the method's originality and relevance to the network inference field is that there is not a direct analysis demonstrating that the novel methodological aspects of BETS have a practical impact. The DREAM4 results are 10 years old and do not reflect the state of the art in Granger causality analysis. Only BETS was run on the glucocorticoid express data. Conceptually, the null distributions and FDR-based approach should improve network quality, but this claim is not directly assessed.

Major comments:

1) Related to the comment above, the manuscript would be improved by more specifically demonstrating to readers why they should use BETS over other modern Granger causality approaches. For instance, if the FDR framework is the main appeal, can it directly demonstrate the advantages of having a principled way to select the size of a network? If it is the scalability and parallelization, can the high-throughput software pipeline be made more robust and user friendly (see below)? There is a comparison with elastic net regression without stability selection, but ensembling or stability selection is now common place in network inference. The most closely related Granger causality work that shares some features with BETS is not included in the DREAM4 benchmarking. Examples of closely related methods include the last two methods in Section 2.1.6 of the supplement (the references are broken so specific manuscripts are unknown) or SWING (Finkle 2018 doi:10.1073/pnas.1710936115), which was not referenced.

2) The analyses all fix the lag at L=2. It is unknown whether that lag would still be appropriate for time series data with more time points. The lag can be set by the user, but the relationships between the lag hyperparameter, the length of the time series, and the effectiveness of the FDR procedures have not been assessed.

3) The permutation and bootstrapping concepts in Figure 1 are understandable at a high level. The methodological details are challenging to follow. It appears that a single temporal permutation is made in the outer loop and another is made in the inner loop. Is a single permutation sufficient to obtain a robust null distribution? In addition, if the bootstrapping is done on the matrix form in Equation 10, which has already unrolled or expanded the L previous time points, how is the inner loop permutation performed? The details of the permutation and FDR procedures are difficult to verify.

4) Using the BETS network's edge signs to predict the vector autoregressive model's edge coefficients is a creative way to use the overexpression data to validate the network that accounts for the edge signs. However, there are some unaddressed caveats with this approach. The overexpression data have fewer time points, so a simpler regression model is used (Equation 17). Any errors in that simple model's fits will confound the BETS network assessment. This approach may also ignore false negative edges in the BETS network. An alternative approach would be to focus on the edge directions in the BETS network by estimating the differentially expressed genes in each overexpression experiment using a temporally-aware statistical test. nsgp (Heinonen 2015 doi:10.1093/bioinformatics/btu699) or a similar method may be able to accommodate the different number of time points in the original and overexpression data. Then, these genes can be treated as the targets of that regulator in a pseudo-gold standard network. The predicted edges for that regular in the BETS network could be evaluated with a precision-recall curve, even if they have very few predicted target genes.

5) The author contributions imply that the RNA-seq data were generated as part of this study. If that is the case, the experimental protocols and methods are incomplete. In addition, the expression data should be made available.

Minor comments:

6) The supplement and GitHub readme note that the time points must be approximately equally spaced. This is an important assumption that limits the applicability to irregular time series. It should be stated more clearly in the main text.

7) The supplement describes the global and local null distributions and FDR approaches in excellent detail. It is difficult to link this discussion to the methods that were actually used in the main text. Stating where the global and local versions were used in the main text methods would help connect these discussions.

8) The main text does not explain the biological goals of the glucocorticoid study or why immune and metabolic genes are of interest. It would help guide readers if some of the well-written explanation from the supplement's Sections 1 and 2.3 was moved to the main text results.

9) The discussion notes that applying Granger causality to single-cell pseudotemporal data is a relevant related area. This is indeed an exciting future direction for BETS, but recent preprints have shown that pseudotimes may not have the same information for network inference as bona fide time series data (Qiu 2018 doi:10.1101/426981; Deshpande 2019 doi:10.1101/534834). That related work may guide readers who attempt to apply BETS to pseudotemporal data.

10) The methods describes BIOGRID PPI, which do not appear to have been used in the analyses described in the results.

11) The references in the supplement are broken

12) The supplement refers to a method named VAR-GEN instead of BETS. Are these the same?

13) Supplemental Figure 7 is difficult to understand. What are the indices and percentages?

14) It is commendable that the software pipelines are available on GitHub with an open source license. Some challenges in running the software are detailed below. In addition, the final version should be archived on Zenodo, Figshare, Software Heritage, or a comparable resource. The Google Drive materials could also be migrated to a permanent repository. The support group https://groups.google.com/forum/#!forum/bets-support displayed an error "You do not have permission to access this content. (#418)".

15) The supplement contains typos "we uses the" (page 9) and "multiple sclrosis patients" (page 29).

Software comments:

The software was tested on Windows 10 with Git for Windows (GNU bash, version 4.4.23(1)-release (x86_64-pc-msys)) in the following Python 2 conda environment:

$ conda create --name bets python=2.7 numpy=1.13 scipy=0.19 pandas=0.20 matplotlib scikit-learn

Python 2's end of life date is in 2020 and support will be dropped by several packages BETS requires (https://python3statement.org/). Porting the code to Python 3 is strongly recommended.

Overall, the software would be much easier to run if it adopted an establishing pipeline workflow. The five major steps require substantial user intervention even though everything could be automated after the options in package_params_cpipeline.sh are configured. This would also help with eventual cross-platform compatibility (currently macOS is supported) and formal testing to ensure the pipeline can run on a different system.

The BETS pipeline in BETS_tutorial.md did not work in the environment described above. After editing dozens of lines, it was possible to run the code through Step 3 (Fit the model on the original data), but then there were too many scripts to edit. The most common incompatibilities were:

- Assuming 'python2' instead of 'python' to run .py files in the shell scripts

- The 'rB' argument when loading pickled files generated "ValueError: Invalid mode ('rB')". Removing the 'rB' worked.

- The 'wB' argument when writing pickled files needed to be 'w' instead.

- The paths to scripts combined different path separators / and \\

- The 'module load' command is not needed to load Python in many systems (this does not terminate execution though)

- Exporting environment variables as a way to configure BETS can be unreliable.

Reviewer #2: In the manuscript, Engelhardt and colleagues describe a new method called Bootstrap Elastic net regression from Time Series (BETS) to infer causal gene networks from time-series gene expression data. BETS uses vector autoregression with elastic net regularization to infer causal relationships (directed edges) between genes. The authors benchmark the performance of BETS by comparing their results against those from 21 other methods on the time-series data from the DREAM4 Network Inference Challenge. Assessed using previously used metrics of performance (AUPR, area under the precision recal curve; AUROC, area under the ROC), BETS’ ranks 6th out of 22 in terms of the AUPR metric (0.13 vs the top-performing CSId @ ~0.2) and ranks 3rd out of 17 in AUROC (0.7 vs the top-performing CSId @0.72). Compared to other top-performing methods, BETS is the fastest (~2-10 times faster), making it an attractive alternative to the top-performing CSId and Jump3. The authors demonstrate the utility of BETS by applying it on a previously published time-series expression (RNA-Seq) data on A549 cells (human adenocarcinoma cell line) exposed to dexamethasone (synthetic glucocorticoid). BETS-inferred causal gene network is validated against orthogonal over-expression datasets.

The proposed method is technically sound, and manuscript is generally well written, concise and easy to read. As the authors correctly state, BETS’ advantage over other methods is its speed, with performance comparable to the best-performing methods. BETS would be a nice addition to the arsenal of methods used to infer causal gene networks from time-series expression data. And, the authors have made their method available on GitHub.

Major Points

1. With regards to the text related to the inferred Glucocorticoid response network (on page 8), where the authors describe the inferred network containing 2,768 nodes (genes): It is noted that all 2,768 genes are ‘effect genes’ (had an incoming directed edge), and 466/2,768 genes are ‘causes’ (causal?), defined as nodes with an outward directed edge. If all the genes have an out-degree (outgoing edge), I don’t understand how the authors can define 466 genes that has both incoming and outgoing edge as causal. I would think that only those nodes with one or more out-going edges and no incoming edge are ‘causal.’ The authors need to explicitly clarify what their definition of ‘causal gene’ is because this raises questions about their ‘causal gene network’ definition.

2. The fact that all 2,768 genes have an out-going edge means that the resulting network/graph is not a directed acyclic graph (DAG; network free of cycles) and that it contains strongly connected component (SCCs), defined as a sub‐networks where, for every pair of nodes u and v in the sub‐network, there exists a directed path from u to v, and from v to u. Given this, I wonder if a method like Vertex Sort (PMID: 19690563) would be more appropriate to infer the network hierarchy and thus ‘causal genes’.

Minor Points:

1. I recommend the authors consider using “time-series” instead of “time series.”

2. The authors have been objective overall in describing/interpreting their results/findings. Line 184 on page 6 states that “BETS had a slightly lower AUPR compared with CSId (~0.12 vs 0.20).” I take issue with ‘slightly’ since CSId’s is ~65% better than BETS w.r.t this metric. I would just remove ‘slightly.’

**Have all data underlying the figures and results presented in the manuscript been provided?**

Reviewer #1: No: The manuscript states "All files have been or are being submitted to the Gene Expression Omnibus under the same study name, including accession number GSE91208." However, that accession number only lists DNase-seq data, not RNA-seq data. Reviewer access links to any private data used in this study should be made available.

Reviewer #2: None

PLOS authors have the option to publish the peer review history of their article (what does this mean?). If published, this will include your full peer review and any attached files.

Reviewer #1: No

Reviewer #2: No

---

## [Decision Letter · Decision Letter 1]

11 May 2020

Dear Dr. Engelhardt,

Thank you very much for submitting your manuscript "Causal network inference from gene transcription time-series response to glucocorticoids" for consideration at PLOS Computational Biology. As with all papers reviewed by the journal, your manuscript was reviewed by members of the editorial board and by several independent reviewers. The reviewers appreciated the attention to an important topic. Based on the reviews, we are likely to accept this manuscript for publication, providing that you modify the manuscript and accompanying software and materials according to the review recommendations.

In particular, Reviewer #1 has concerns about the usability of the BETS software, as this reviewer tried and was not able to run the package.  Minor additional concerns include ensuring the availability of the overexpression data by depositing in GEO as well as the source code and supplementary materials by placing in a suitable repository.

Sincerely,

Christina S. Leslie

Associate Editor

PLOS Computational Biology

Thomas Lengauer

Methods Editor

PLOS Computational Biology

[LINK]

Reviewer's Responses to Questions

**Comments to the Authors:**

Reviewer #1: The authors have made substantial revisions that greatly enhance the manuscript and address almost all of my comments with the initial submission. These include clarifications in the Methods text, demonstration of robustness to the lag parameter values, results from new network inference algorithms, and another approach for evaluating the glucocorticoid predictions. Even though not all of the quantitative results favor the BETS algorithm, the additional results strengthen the manuscript. The authors have conducted fair and objective evaluations instead of skewing the results and language to only highlight advantages of BETS. For instance, they are honest in reporting that SWING-RF is now the top performer in the DREAM dataset and the nsgp-based glucocorticoid evaluation shows the BETS predictions are not significantly better than random guessing. (Those glucocorticoid predictions are still supported by two other analyses.) All network inference algorithms have some limitations, so the objective assessment will raise readers' confidence in the reported results and improves the manuscript overall.

The main reason I am enthusiastic about this manuscript is that the false discovery rate framework is a major and novel contribution to the network inference field. Figure 2 shows that BETS substantially improves upon other vector autoregressive network inference algorithms (including the new SWING-Lasso) in the DREAM evaluation. BETS remains very good at identifying trans-eQTLs. In addition, the analyses and results are rigorous, and the manuscript is very polished overall.

The only remaining major concern is the usability of the BETS software, as detailed below.

Major comments:

1) I am still unable to run the BETS pipeline. This time I tried running BETS in Python 3 on a Linux server. I created a fresh conda environment with

$ conda create --name bets python=3 numpy=1.13 scipy=0.19 pandas=0.20 matplotlib scikit-learn

There were fatal errors within the first few minutes of running the pipeline described in BETS_tutorial.md

- The line 'export NULL=g' comes after $NULL is used in package_params_cpipeline.sh so the directory names are incorrect

- prep_jobs_bootstrap.sh still uses the 'python2' command to run the Python script so it does not work in a Python 3 environment. Because this command is run within a script, aliasing python2 to point to python3 does not work.

Because the software is an important part of the contribution, the authors should ensure the pipeline can run in a fresh Python environment. One way to guarantee this would be to run it in a continuous integration service like Travis CI or GitHub Actions.

Minor comments:

2) To keep the discussion balanced, some of the new negative results could be included alongside the existing positive results. SWING-RF is very fast and accurate (Table S3), but it is not included in the runtime discussion on line 191. BETS is substantially faster than CSId and Jump3 but not SWING-RF. The Discussion paragraph starting at line 379 only focuses on the positive validations for the glucocorticoid network but ignores the nsgp-based results

3) The editors should ensure the overexpression data is available on GEO before publication.

4) The source code and supplementary materials on Google Drive should be archived in a more permanent repository, even if they are > 100 GB. The NIH figshare instance allows 100 GB of storage, and researchers can request more (https://nih.figshare.com/f/faq). Zenodo has 50 GB by default, but more is available upon request (https://about.zenodo.org/policies/). PLOS Computational Biology partners with Dryad (https://datadryad.org/stash/publishing_charges) and offers a 300 GB limit (https://datadryad.org/stash/faq)

5) Some inconsistencies and typos have been introduced over the different versions of the manuscript

- References to 'STAR Methods' remain

- Some text describes 340 trans-eQTLs, other text states 341

- Line 442 states 'In BETS, L = 2.' but now the results include multiple values of L, so this could state L = 2 is the default

- Line 476 'that the the'

- Line 576 states 'reference numbers listed in Supplementary Table 3' but that table contains other data

- The 'DREAM benchmarking' section (line 601) omits the new SWING methods

- Line 638 still refers to STRING interactions

- Line 799 'gene g \\in \\not tf a random score' is missing a word

Reviewer #2: The authors have have satisfactorily addressed the reviewers' comments.

**Have all data underlying the figures and results presented in the manuscript been provided?**

Reviewer #1: No: Not yet. The accession number GSE91208 is listed but does not seem relevant. The GEO accession numbers for the 100 nM dexamethasone treatment time course have now been provided. The GEO uploads for the overexpression data are still in progress. The Google Drive data should still be archived as well.

Reviewer #2: Yes

PLOS authors have the option to publish the peer review history of their article (what does this mean?). If published, this will include your full peer review and any attached files.

Reviewer #1: No

Reviewer #2: No
---

## [Decision Letter · Decision Letter 2]

7 Aug 2020

Dear Dr. Engelhardt,

We are pleased to inform you that your manuscript 'Causal network inference from gene transcription time-series response to glucocorticoids' has been provisionally accepted for publication in PLOS Computational Biology. Please note the reviewers additional comments, though.

Best regards,

Christina S. Leslie

Associate Editor

PLOS Computational Biology

Thomas Lengauer

Methods Editor

PLOS Computational Biology

Reviewer's Responses to Questions

**Comments to the Authors:**

Reviewer #1: The authors have addressed my previous comments about the manuscript, data, and software. I was able to access the GEO datasets and spot checked the expression data. The supplementary results and files are now archived on Zenodo. Finally, I confirmed that I could run BETS on a Linux server in the conda environment described in the previous review. I have no other concerns and remain enthusiastic about this research.

One last comment is that the Zenodo file port-from-della.zip contains many files that could be removed. For instance, I noticed

- drive-download-20200629T235020Z-001.zip

- ProbGenReceipt_2016.pdf

- Goldwater_ResearchEssay_1_25_17.docx

- BACKUP* files

- The code/ subdirectory

- Many other files in the presentations/ subdirectory that aren't referenced in 'Full Progeny.xlsx'

The Zenodo dataset can be updated at any time by uploading a new version of the zip file, so this suggestion does not impact my recommendation for the manuscript.

**Have all data underlying the figures and results presented in the manuscript been provided?**

Reviewer #1: Yes

PLOS authors have the option to publish the peer review history of their article (what does this mean?). If published, this will include your full peer review and any attached files.

Reviewer #1: No

---

## [Editor Report · Acceptance letter]

22 Jan 2021

PCOMPBIOL-D-19-01120R2 

Causal network inference from gene transcription time-series response to glucocorticoids

Dear Dr Engelhardt,

I am pleased to inform you that your manuscript has been formally accepted for publication in PLOS Computational Biology. Your manuscript is now with our production department and you will be notified of the publication date in due course.

With kind regards,

Jutka Oroszlan
